# Metabolic mutations reduce antibiotic susceptibility of *E. coli* by pathway-specific bottlenecks

Paul Lubrano[1,2,3,9], Fabian Smollich [1,2,3,9], Thorben Schramm[1,8], Elisabeth Lorenz[1], Alejandra Alvarado [1,2], Seraina Carmen Eigenmann[4], Amelie Stadelmann[1,2,3], Sevvalli Thavapalan [1,2,3], Nils Waffenschmidt[1], Timo Glatter[5], Nadine Hoffmann[2,6], Jennifer Müller [6,7], Silke Peter[2,6,7], Knut Drescher [4] & Hannes Link [1,2,3]✉

## Abstract

Metabolic variation across pathogenic bacterial strains can impact their susceptibility to antibiotics and promote the evolution of antimicrobial resistance (AMR). However, little is known about how metabolic mutations influence metabolism and which pathways contribute to antibiotic susceptibility. Here, we measured the antibiotic susceptibility of 15,120 *Escherichia coli* mutants, each with a single amino acid change in one of 346 essential proteins. Across all mutants, we observed modest increases of the minimal inhibitory concentration (twofold to tenfold) without any cases of major resistance. Most mutants that showed reduced susceptibility to either of the two tested antibiotics carried mutations in metabolic genes. The effect of metabolic mutations on antibiotic susceptibility was antibiotic- and pathway-specific: mutations that reduced susceptibility against the β-lactam antibiotic carbenicillin converged on purine nucleotide biosynthesis, those against the aminoglycoside gentamicin converged on the respiratory chain. In addition, metabolic mutations conferred tolerance to carbenicillin by reducing growth rates. These results, along with evidence that metabolic bottlenecks are common among clinical *E. coli* isolates, highlight the contribution of metabolic mutations for AMR.

**Keywords** Metabolism; Antibiotic Resistance; Mutations; *Escherichia coli*; CRISPR
**Subject Categories** Metabolism; Microbiology, Virology & Host Pathogen Interaction

## Introduction

Antimicrobial resistance (AMR) is a major threat to global health, and the associated death toll is alarmingly increasing (Mattar et al, 2020). Among the major contributors to AMR are pathogenic strains of *Escherichia coli*, which have been associated with more than 800,000 deaths in 2019 (Murray et al, 2022). AMR is either a consequence of mobilized resistance genes (e.g., drug-modifying enzymes), or of mutations that change drug transport or binding to the drug–target (Yelin and Kishony, 2018; Darby et al, 2023). Apart from such canonical resistance mechanisms, mutations in genes that are not directly related to the drug or the drug–target can also confer antibiotic resistance or promote its evolution. However, these noncanonical resistance mutations are difficult to identify due to their indirect effects on antibiotic action, which is often mediated by changes in bacterial physiology and metabolism (Stokes et al, 2019; Brauner and Balaban, 2021; Bhargava and Collins, 2015). For example, mutations in arginine biosynthesis genes of *Mycobacterium smegmatis* upregulated an aminoglycoside-modifying enzyme (Schrader et al, 2021), and a hypomorphic variant of an enzyme in $CO_2$ metabolism promoted fluoroquinolone resistance in *Neisseria gonorrhoeae* (Rubin et al, 2023). Laboratory evolution studies have provided further evidence for the role of metabolism in antibiotic resistance, suggesting that mutations in metabolic genes have clinical relevance (Lopatkin et al, 2021), and that they influence the evolutionary pathway towards resistance (Zampieri et al, 2017).

Despite first approaches to map antibiotic resistance of single genes with single-nucleotide resolution (Garst et al, 2017; Dewachter et al, 2023), it remains difficult to delineate cellular functions and metabolic pathways that are most relevant for AMR. Metabolic pathways that are increasingly associated with antibiotic action are purine nucleotide metabolism (Lopatkin and Yang, 2021) and respiration (Lobritz et al, 2015; Van den Bergh et al, 2022). For example, ATP levels have been associated with antibiotic tolerance in *E. coli* (Lopatkin et al, 2019; Yang et al, 2019), as well as with persister formation in *Staphylococcus aureus* (Conlon et al, 2016) and *E. coli* (Wilmaerts et al, 2018; Shan et al, 2017). While these studies demonstrated the role of metabolism in antibiotic lethality, mutations in metabolic genes that confer antibiotic resistance are rarely identified. This is because most studies tend to focus on

[1]Interfaculty Institute of Microbiology and Infection Medicine, University of Tübingen, Auf der Morgenstelle 24, 72076 Tübingen, Germany. [2]Cluster of Excellence "Controlling Microbes to Fight Infections", University of Tübingen, 72076 Tübingen, Germany. [3]M3 Research Center, Otfried-Müller-Straße 37, University of Tübingen, 72076 Tübingen, Germany. [4]Biozentrum, University of Basel, Spitalstrasse 41, 4056 Basel, Switzerland. [5]Max Planck Institute for Terrestrial Microbiology, Karl-von-Frisch-Straße 10, 35043 Marburg, Germany. [6]Institute of Medical Microbiology and Hygiene, University of Tübingen, Elfriede-Aulhorn-Str. 6, 72076 Tübingen, Germany. [7]NGS Competence Center Tübingen (NCCT), 72076 Tübingen, Germany. [8]Present address: Institute of Molecular Systems Biology, ETH Zurich, Otto-Stern-Weg 3, 8093 Zürich, Switzerland. [9]These authors contributed equally: Paul Lubrano, Fabian Smollich. ✉E-mail: hannes.link@uni-tuebingen.de

single isolates and individual mutations (Schrader et al, 2021; Rubin et al, 2023), with few systematic analyses that explore the full spectrum of resistance mutations across metabolic genes (Lopatkin et al, 2021).

Antibiotic resistance and antibiotic tolerance are two distinct mechanisms by which bacteria evade the effects of antibiotics (Brauner et al, 2016). Resistance is the inherited ability of bacteria to grow at high antibiotic concentrations. Tolerance, in contrast, is the ability to survive exposure to high concentrations of an antibiotic without a corresponding increase in minimum inhibitory concentration (MIC). Tolerance is often linked to slowing down of essential bacterial processes (Brauner et al, 2016), but how essential processes influence antibiotic resistance remains unclear.

The aim of this study was to test if slowing down essential processes confers resistance to antibiotics and to systematically identify these processes. Therefore, we used a CRISPR library of *E. coli* strains, each carrying a point mutation in an essential gene, many of which are likely to reduce their activity. By measuring the antibiotic susceptibility of these strains, we sought to determine whether mutations in essential genes can increase the MIC of antibiotics. Across 15,120 *E. coli* mutants (each with a single amino acid change in one of 346 essential proteins) we observed twofold to tenfold MIC increases, but no case of major resistance. Although these mutants do not meet the definition of clinical resistance, because they do not grow above the clinical breakpoint concentrations when tested under standard conditions, we will refer to them as resistance mutations for simplicity in the following. Most mutations that conferred resistance to the β-lactam antibiotic carbenicillin occurred in genes that are involved in purine nucleotide and amino acid biosynthesis. Resistance mutations against gentamicin occurred in metabolic pathways related to the respiratory chain. These results demonstrate that metabolic mutations confer antibiotic resistance in a pathway-specific manner, rather than by a general reduction in growth rate or an overall metabolic state. However, while the same metabolic mutations that conferred resistance to carbenicillin also led to antibiotic tolerance, this tolerance was mainly a result of the reduced growth rates of the metabolic mutants, which is consistent with previous studies (Lee et al, 2018). Finally, we analyzed growth and metabolism of clinical *E. coli* isolates and identified metabolic bottlenecks similar to those observed in our CRISPR mutants.

# Results

## Mapping mutations in essential genes that confer antibiotic resistance

We hypothesized that hypomorphic (or partial loss-of-function) mutations in essential genes confer antibiotic resistance because they decrease cellular growth and metabolic activities. To test this hypothesis, we measured the antibiotic resistance of an *E. coli* CRISPR library that we constructed in a previous study (Schramm et al, 2023). In that study, we selected 352 proteins that are essential for *E. coli* growth on minimal glucose medium. For each protein, we designed up to 50 amino acid changes (10 sites, with 5 substitutions per site). For 154 proteins, fewer than 50 substitutions were feasible due to limitations by our design rules for CRISPR gene editing. The mutations were designed to destabilize

the proteins and were primarily located at buried sites. After gene editing, the library contained 15,120 of the 16,038 designed single amino acid substitutions and targeted 346 genes. Most mutations (8290) were classified as "low-fitness" mutations (Schramm et al, 2023) and these mutations are the focus of this study, because they may slow down the associated essential processes.

Each strain in the library carries two plasmids (Fig. 1A). The first plasmid (pTS040) has a repair template and a strain-specific sgRNA for gene editing, and a chloramphenicol resistance marker for selection. The second plasmid (pTS041) has the genes encoding the lambda-derived Red system (Exo, Beta, Gam) and the *Streptococcus pyogenes cas9* gene, with a kanamycin resistance marker. To avoid any bias from the antibiotics used for plasmid selection (kanamycin and chloramphenicol), we used a control strain containing the same plasmids (pTS040 and pTS041) and cultivated it under the same conditions as the CRISPR library, where kanamycin and chloramphenicol were added to all cultures. The repair template is used only once to introduce the mutation during the gene editing step. Afterward, the genomic mutation remains stable in the *E. coli* cells, which we confirmed by sequencing of genomic DNA of selected mutants. Screening of all mutants in the library relies on sequencing of the repair template and the sgRNA on pTS040, which serve as a strain-specific barcode (Fig. 1A).

First, we grew the CRISPR library and the control strain on agar plates containing 2×, 4×, 6×, and 8× the MIC of carbenicillin (a β-lactam) and gentamicin (an aminoglycoside). For carbenicillin, we observed a marked difference in colony-forming units (CFUs) between the control strain and the library at 2× MIC, but this difference disappeared at 4× MIC and above (Appendix Fig. S1). This suggests that most mutations in the library confer only low-level resistance to carbenicillin. For gentamicin, CFU differences between the CRISPR library and the control strain were still noticeable up to 8× MIC (Appendix Fig. S1), but we used 2× MIC for both antibiotics in the subsequent screen for resistant mutants.

Next, we screened the CRISPR library against carbenicillin and gentamicin at 2× MIC of the control strain (Fig. 1A). As a reference, we cultivated the library on agar plates without carbenicillin or gentamicin (reference plate, Fig. 1A). Each experiment was performed twice on different days to test reproducibility. After 48 h of incubation at 2× MIC, markedly more colonies formed on antibiotic plates inoculated with the CRISPR library compared to antibiotic plates inoculated with the control strain (Appendix Fig. S2). To identify resistant mutants in the CRISPR library, we harvested the colonies from agar plates and determined read counts of strain-specific barcodes by deep sequencing, which was reproducible between the two replicates (Fig. EV1). Fold changes for each mutant were calculated as the ratio of barcode read fractions on antibiotic plates relative to reference plates (Fig. 1B). Mutants with fold-changes >20 in both replicates were considered putatively resistant, resulting in 149 resistance mutations for carbenicillin, and 83 resistance mutations for gentamicin (Dataset EV1).

For carbenicillin, 123 of the 149 identified resistance mutations were located in genes that had multiple resistance mutations. For example, 27 genes had more than one resistance mutation, and the most frequently mutated genes were *purM* and *purD* in the purine nucleotide biosynthesis pathway (Fig. 1C). The repeated occurrence

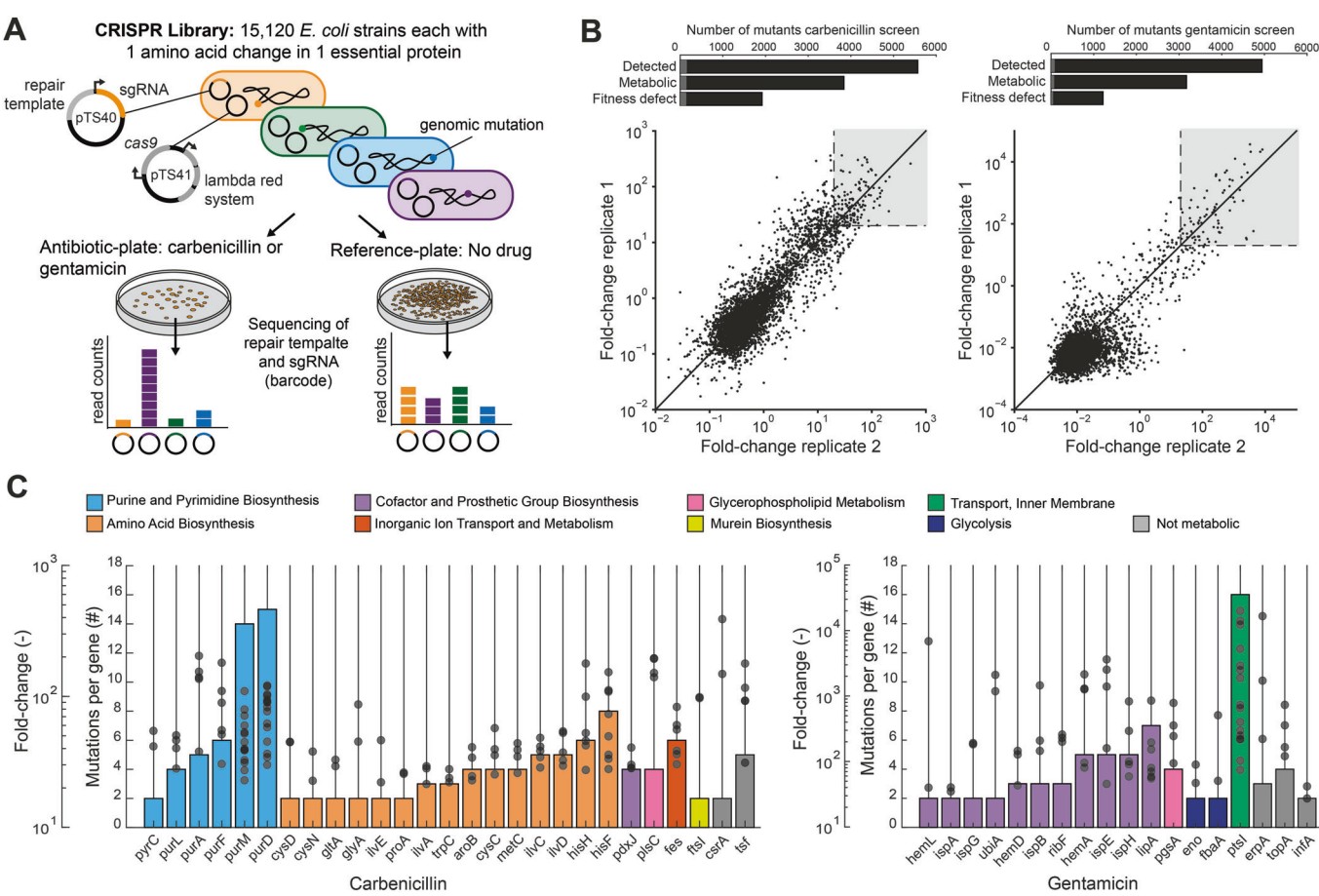

**Figure 1. Mapping antibiotic resistance mutations with 15,120 *E. coli* mutants.**

(A) Schematic of the CRISPR screen. The CRISPR library included 15,120 *E. coli* strains, each with a single amino acid change in one of 346 essential proteins. Each mutant has a sgRNA and a repair template on a plasmid. The CRISPR library was cultivated on agar plates with and without the antibiotic ($n = 2$ replicates at different days). Antibiotics were added at 2× of the minimal inhibitory concentration (MIC). Strain-specific barcodes (sgRNA and repair template) were sequenced after 48 h of incubation to determine the composition of the library. (B) Fold change of single mutants in the CRISPR library on carbenicillin and gentamicin. Fold changes were calculated as barcode read fractions on antibiotic plates relative to barcode read fractions on the reference plates. Strains with a fold change >20 in both replicates are considered putatively resistant against the respective antibiotic (gray regions). Bar plots show the number of mutants that: were detected in the screen (Detected), occurred in metabolic genes (Metabolic), and had a fitness defect based on our previous screen with the library (Schramm et al, 2023) (Fitness defect). Gray bars are mutants with fold change >20. (C) Genes with a putative resistance mutation against gentamicin (right) or carbenicillin (left). Bars show the number of different resistance mutations per gene. Dots show the mean fold change of each mutation. Genes are grouped by functional categories in the genome-scale metabolic model of *E. coli* iML1515 (Monk et al, 2017). Amino acid metabolism categories are shown as one functional category. Source data are available online for this figure.

of resistance mutations in the same gene was a first indication that these are bona fide resistance mutations. In the case of gentamicin, 72 of the 83 identified resistance mutations were located in genes that had multiple resistance mutations, and 18 genes had more than one resistance mutation (Fig. 1C). Most mutations occurred in *ptsI*, a component of the phosphoenolpyruvate:carbohydrate phosphotransferase system (PTS) that transports and phosphorylates glucose in *E. coli*.

In summary, our CRISPR library included *E. coli* mutants with single amino acid changes that confer resistance to carbenicillin and gentamicin. Importantly, resistance mutations occurred mostly in metabolic genes (95% on carbenicillin and 84% on gentamicin), although 31% of all mutations detected in our screen were not metabolic (Fig. 1B). This indicates that cellular metabolism impacts carbenicillin and gentamicin resistance in *E. coli*, although this effect

seemed to be associated with modest MIC increases (Appendix Fig. S1). To further understand the range of MIC increases, we sequenced 96 individual colonies from plates with 6X MIC of carbenicillin, as well as 96 colonies from plates with 10× MIC of gentamicin (Dataset EV2). For carbenicillin, only 12 out of 96 strains matched the library barcodes, whereas 51 out of 96 strains matched for gentamicin. Most gentamicin-resistant mutations overlapped with those identified at 2× MIC, suggesting that metabolic mutations can confer higher resistance levels. In contrast, for carbenicillin, only one mutation, HisA[V224Q], overlapped with the 2× MIC mutants, indicating that achieving higher resistance to carbenicillin via mutations in our library is difficult. These results are consistent with the CFU counts at higher MICs for the CRISPR library and the control strain (Appendix Fig. S1). Next, we examined the pathway-specificity of resistance mutations.

## Resistance by metabolic mutations is caused by pathway-specific effects and not by fitness defects

Our CRISPR screen identified resistance mutations that mostly affected metabolic genes, and we mapped them to the functional categories of the *E. coli* metabolic model *i*ML1515 (Monk et al, 2017) (Fig. 1C; Dataset EV1). For gentamicin, most mutations occurred in cofactor and prosthetic group biosynthesis pathways that produce components of the respiratory chain, such as genes in the biosynthesis of heme (e.g., *hemA*) and ubiquinone (*ubiA*). Aminoglycosides bind ionically to the outer membrane and utilize the respiratory chain for cytosolic entry in *E. coli* (Taber et al, 1987). Previous studies showed that mutations in the respiratory chain-related genes *ubiF* and *hemA* lead to low-level aminoglycoside resistance due to decreased drug uptake (Muir et al, 1981; Bryan and Van Den Elzen, 1977). Mutations in phosphatidylglycerophosphate synthase (*pgsA*) have been associated with tobramycin resistance (Pal and Andersson, 2024) and might also alter the respiratory chain since cardiolipin acts as a proton trap (Haines and Dencher, 2002). The CRISPR screen identified seven genes with gentamicin-resistance mutations that were not metabolic (Dataset EV1), including DNA topoisomerase *topA* (4 mutations).

For carbenicillin, 44 resistance mutations affected metabolic enzymes in de novo biosynthesis of purine nucleotides. For example, 5 mutations occurred in *purA* (adenylosuccinate synthetase) and 4 of them had high fold changes (>100). Another 54 mutations affected genes in amino acid biosynthesis, with 14 mutations in the histidine biosynthesis enzyme imidazole glycerol phosphate synthase (encoded by *hisF* and *hisH*). Although the heterodimer HisFH is an enzyme of the histidine biosynthesis pathway, it also produces the purine intermediate 5-amino-1-(5-phospho-D-ribosyl)imidazole-4-carboxamide (aicar). Thus, we assumed that mutations in *hisF* and *hisH* perturbed both the histidine and the purine pathway. Two resistance mutations occurred in the peptidoglycan DD-transpeptidase-encoding gene *ftsI*, which is a direct target of β-lactams (Miller et al, 2004), and the mutations may reduce carbenicillin binding. The carbenicillin screen identified only three genes that were not annotated to the *E. coli* metabolic model: the arginine—tRNA ligase *argS* (1 mutation), the elongation factor *tsf* (5 mutations) and the carbon storage regulator *csrA* (2 mutations). Since deletion of the transcription factor *csrA* has been shown to decrease expression of the porin OmpF in *E. coli*, the two *csrA* mutations observed in our screen might affect carbenicillin transport (Potts et al, 2017). However, to our knowledge, *csrA* has not yet been associated with β-lactam resistance.

Thus, our CRISPR screen identified potential resistance mutations, with a significant enrichment of mutations in metabolic genes compared to non-metabolic genes (Table EV1). A large fraction of these genes was in purine and amino acid biosynthesis for carbenicillin, and in the respiratory chain for gentamicin, indicating that resistance to these antibiotics results from specific metabolic perturbations rather than a universal metabolic state or general growth defects, which have previously been linked to antibiotic lethality (Lopatkin et al, 2019; Lee et al, 2018). Further, comparing the resistance phenotype of each mutant with their fitness phenotypes indicated that a growth defect alone does not confer resistance, because most mutants with a fitness defect did not show resistance in the screen against either antibiotic (Fig. 1B;

Dataset EV1). This implies that the putative resistance phenotypes are pathway-specific and that reduced growth alone is not sufficient to confer resistance.

## Metabolic mutations induce bottlenecks in their associated pathways

To validate putative resistance mutations from the CRIPSR screen, we re-constructed three carbenicillin-resistant mutants: PurM$^{F105A}$ in the upper purine pathway, PurA$^{L75D}$ in the ATP branch, and HisF$^{V126P}$ at the branchpoint between the histidine and the purine pathway (Fig. 2A). All three mutants were resistant to carbenicillin, with at least twofold higher MICs than the non-edited control strain (Fig. 2B). The purine mutants were also resistant against aztreonam but not to meropenem (Appendix Fig. S3), suggesting that the resistance mechanism is linked to the structure of the respective β-lactam, because the β-lactam structure influences drug transport (Prajapati et al, 2021) or penicillin-binding protein specificity (Spratt, 1977). Additionally, we confirmed that the mutants PtsI$^{I330P}$, HemA$^{L276Q}$, RibD$^{L364W}$, PgsA$^{V44P}$, and IspE$^{V146W}$ (Fig. 2C) are resistant against gentamicin (Fig. 2D), as well as tobramycin, another aminoglycoside (Appendix Fig. S3). To test the potential combination effects of carbenicillin or gentamicin with the selection markers kanamycin and chloramphenicol, we made agar dilution assays on plates without the selection markers for the PurA$^{L75D}$ and HemA$^{L276Q}$ mutants. Both mutants showed resistance in the absence of kanamycin and chloramphenicol (Appendix Fig. S4), but the susceptibility towards gentamicin was slightly increased for both the control and HemA$^{L276Q}$ strain.

Resistance was tested with agar dilution assays using several inoculum densities (Wiegand et al, 2008). In some cases, we found an inoculum effect where cells grew at high inoculum density (0.1 OD$_{600}$), and showed no growth or only single colonies at low inoculum density (0.0001 OD$_{600}$). This behavior could be caused by an artificial reduction of antibiotic concentration or because cells grow on a layer of dead cells. Therefore, we based our MIC determination only on the spots at the lower inoculum density (0.0001 OD$_{600}$).

To understand how metabolic mutations influence cellular metabolism, we quantified intracellular metabolites with targeted metabolomics. Therefore, we grew the mutants on minimal glucose medium without antibiotics and measured metabolites during exponential growth with liquid chromatography-tandem mass spectrometry (LC-MS/MS) (Guder et al, 2017). In the carbenicillin-resistant mutants PurA$^{L75D}$ and PurM$^{F105A}$, stronger metabolic changes occurred in nucleotide biosynthesis with decreases between twofold and fivefold in the purine nucleotide end-products ATP and GTP (Fig. 2E). GTP levels were less perturbed in the PurA$^{L75D}$ strain, probably because PurA catalyzes the first step in the ATP branch of the purine pathway. The strongest metabolome changes in the PurA$^{L75D}$ strain and PurM$^{F105A}$ strain were increases of the substrate metabolites of PurA and PurM, inosine monophosphate (imp) and 2-(formamido)-N1-(5-phospho-β-D-ribosyl)acetamidine (fpram), respectively. These substrate increases indicate that PurA$^{L75D}$ and PurM$^{F105A}$ are hypomorphic mutations that decrease the catalytic capacity of the enzymes, which in turn limits de novo biosynthesis of purine nucleotides. The HisF$^{V126P}$ strain had also low levels of purine nucleotides, as well as low histidinol (histd) levels, thus indicating a

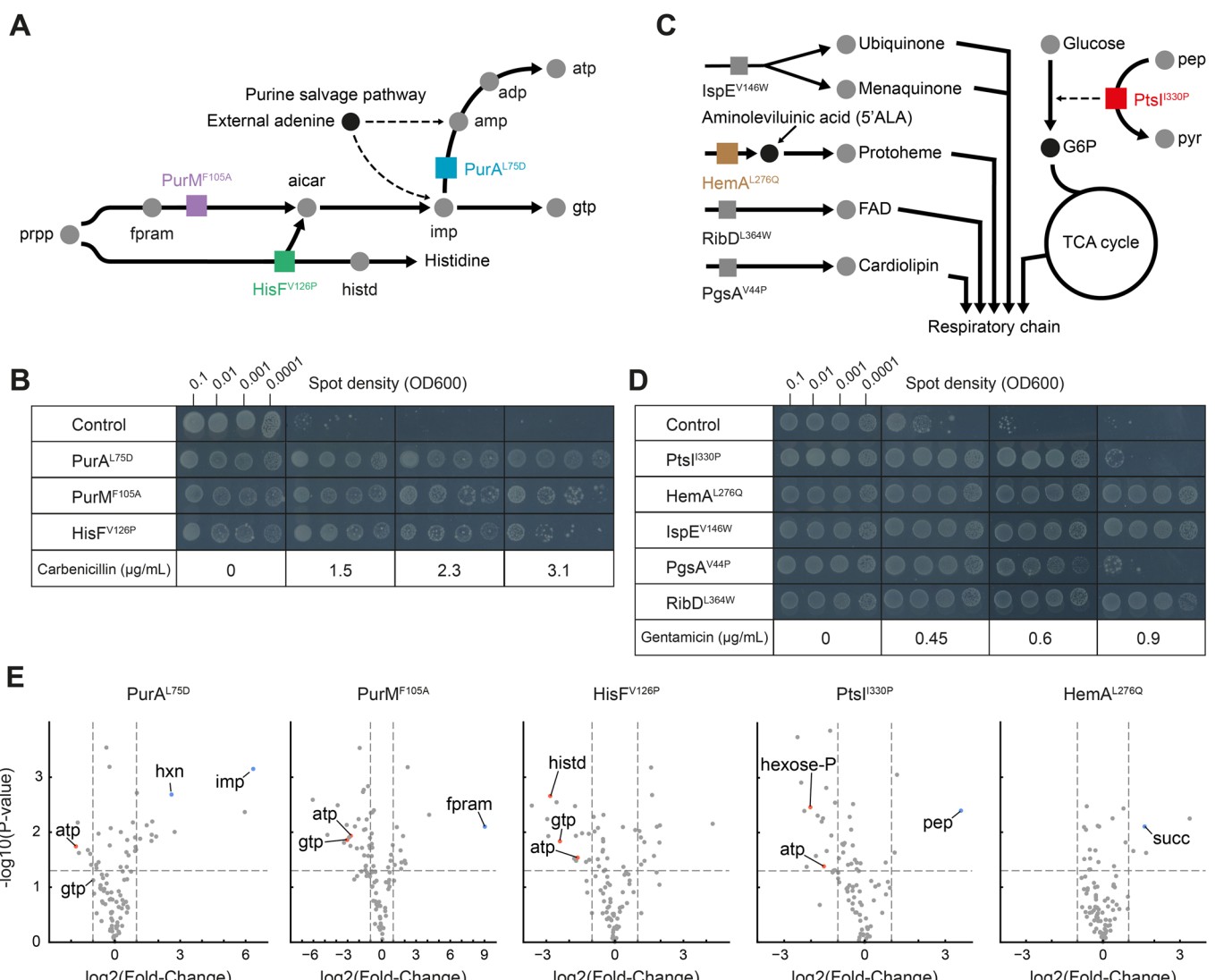

**Figure 2. Metabolic mutations lead to biosynthetic bottlenecks that induce antibiotic resistance.**

(A) Schematic of the biosynthesis pathways of purine nucleotides and histidine. Colored boxes indicate the enzymes with carbenicillin resistance mutations: HisF[V126P], PurM[F105A], and PurA[L75D]. (B) Agar dilution assay with the control strain and three re-constructed mutants (HisF[V126P], PurM[F105A] and PurA[L75D]). Each strain was spotted on agar plates with minimal glucose medium containing increasing concentrations of carbenicillin (control strain MIC = 1.5 µg/mL). Multiple inoculum densities were used to assess inoculum effects. Plates were incubated 48 h. Shown is one of n = 2 replicates. (C) Schematic of multiple metabolic pathways that are associated to gentamicin-resistance mutations: PtsI[I330P], HemA[L276Q], IspE[V146W], PgsA[V44P], RibD[L364W]. All pathways converge towards the respiratory chain. Boxes represent mutated enzymes with resistance mutations. (D) Agar dilution assay with the control strain and mutants that were identified in the gentamicin screen (PtsI[I330P], HemA[L276Q], IspE[V146W], PgsA[V44P], RibD[L364W]). Each strain was plated on agar plates with minimal glucose medium containing increasing concentrations of gentamicin (control strain MIC = 0.45 µg/mL). Multiple inoculum densities were used to assess inoculum effects. Plates were incubated 48 h. Shown is one of n = 2 replicates. (E) Volcano plots show metabolite levels of resistant mutants as fold changes relative to the control strain (n = 3 replicates). Significant metabolite changes (P value < 0.05, FC > 1, two-tailed t test) that are discussed in the text are annotated. Source data are available online for this figure.

deficiency of the HisFH complex that leads to bottlenecks in both the lower histidine pathway and the purine pathway.

In the PurA[L75D] mutant, we observed high levels of inosine and hypoxanthine (Fig. 2E), probably due to the accumulation of the PurA substrate IMP. Since hypoxanthine is an allosteric activator of the purine repressor PurR, we expected a downregulation of purine biosynthesis genes. Consistent with this, the transcriptome of the PurA[L75D] strain showed a significant downregulation of genes in purine nucleotide biosynthesis, which may aggravate the

bottleneck in the pathway (Appendix Fig. S5; Dataset EV3). The strongest transcriptome response in the PurA[L75D] strain was a downregulation of the xanthine transporter *xanP* and an upregulation of the pyruvate transporter *yjiY*. Moreover, genes associated with acid stress response were upregulated (e.g., *gadABC* and *hdeABD*). These transcriptional changes, individually or in combination, may contribute to the MIC increases in the PurA[L75D] strain, potentially by altering transport processes or stress responses.

The metabolome of gentamicin mutants indicated that they also introduce bottlenecks in their associated pathways. For example, the strongest metabolite change in the PtsI$^{I330P}$ strain was an increase of phosphoenolpyruvate (pep), a substrate metabolite of the PTS (Fig. 2E). This indicated a metabolic bottleneck at the initial step of glycolysis, which is further supported by low levels of hexose phosphates (hexose-p) in the PtsI$^{I330P}$ strain. Although an *E. coli* mutant lacking PtsI ($\Delta ptsI$) has been linked to antibiotic tolerance (Zeng et al, 2022), a role in antibiotic resistance has not been reported to our knowledge. In the HemA$^{L276Q}$ and RibD$^{L364W}$ mutants, we detected high levels of succinate (succ, Fig. 2E; Appendix Fig. S6). Succinate is the substrate metabolite of succinate dehydrogenase (Sdh), which requires heme b groups as well as FAD as cofactor. Therefore, it is likely that the HemA$^{L276Q}$ and RibD$^{L364W}$ mutants have metabolic bottlenecks that respectively limit the production of heme and FAD, which leads to a bottleneck at Sdh and blocks the respiratory chain.

Thus, metabolic mutations can lead to bottlenecks in their associated pathways and may induce antibiotic resistance. To obtain additional evidence that resistance was due to the bottlenecks and not pleiotropic effects caused by genome editing, we used CRISPR interference (CRISPRi) to decrease the concentration of PurA. The CRISPRi-*purA* strain was indeed resistant to carbenicillin and its metabolome was similar to the PurA$^{L75D}$ strain metabolome, with strong increases of the PurA substrate IMP (Appendix Fig. S7).

If a limited supply of specific metabolites induced antibiotic resistance in our mutants, we expected that external sources of these metabolites would reverse resistance. *E. coli* can convert external adenine into ATP and GTP by nucleotide salvage pathways that are independent from de novo biosynthesis (Xi et al, 2000) (Fig. 2A). As expected, feeding adenine to the PurA$^{L75D}$, PurM$^{F105A}$, and HisF$^{V126P}$ strains reversed their carbenicillin resistance

(Fig. EV2). We also confirmed that adenine supplementation restored the purine nucleotide levels in the PurA$^{L75D}$ strain to the levels of the control strain (Appendix Fig. S8). Similarly, the PtsI$^{I330P}$ mutant was not resistant to gentamicin during growth on glucose-6-phosphate, a non-PTS sugar (Fig. EV3), and supplementing a heme biosynthesis intermediate (5-aminolevulinic acid) reversed gentamicin resistance of the HemA$^{L276Q}$ mutant (Fig. EV3).

In summary, metabolome analyses demonstrated that metabolic mutations induce specific bottlenecks in their associated pathways, which contribute to antibiotic resistance. This resistance can be reversed by supplementing external sources of the limiting metabolites, confirming that the resistance mechanisms are linked to bottlenecks within specific metabolic pathways. Next, we investigated if resistance was only due to pathway-specific effects or if global growth defects also play a role in resistance.

## Growth defects from metabolic mutations influence tolerance but not resistance

To investigate the influence of cellular growth on resistance, we assessed the growth rates of the three carbenicillin-resistant mutants (PurM$^{F105A}$, PurA$^{L75D}$ and HisF$^{V126P}$) and five gentamicin-resistant mutants (PtsI$^{I330P}$, HemA$^{L276Q}$, RibD$^{L364W}$, PgsA$^{V44P}$ and IspE$^{V146W}$) in minimal glucose medium (Fig. 3A). All mutants had significantly lower growth rates than the control strain (*P* value < 0.05). Given the known associations between growth rates and antibiotic efficacy (Lee et al, 2018), we explored whether slow growth contributes to resistance of the mutants. To account for such an effect, we used a LeuB$^{I134P}$ mutant, which had a low growth rate (Fig. 3A) and was not resistant to carbenicillin (Appendix Fig. S9), thus indicating that slow growth alone does not confer resistance. This was further supported by testing cross-resistance of the gentamicin and carbenicillin-resistant mutants: the slow-

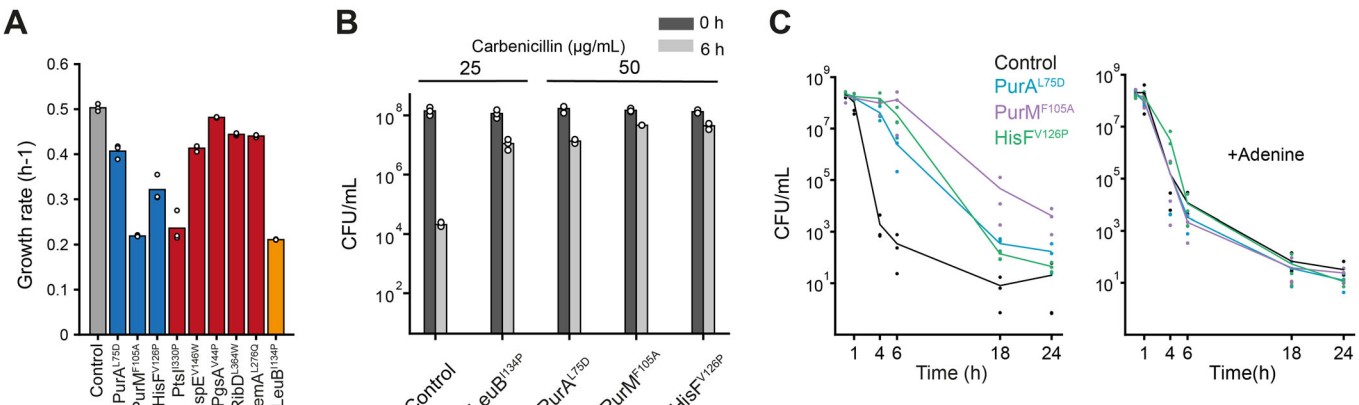

**Figure 3. Slow growth but not metabolic bottlenecks confer β-lactam tolerance.**

(**A**) Growth rates of control strain and selected mutants in minimal medium with glucose. Blue: carbenicillin-resistant mutants. Red: gentamicin-resistant mutants. Orange: slow growth control strain LeuB$^{I134P}$ (*n* = 3 replicates). (**B**) Survival of carbenicillin treatment of the control strain, the LeuB$^{I134P}$ mutant (slow growth control) and three purine mutants (PurA$^{L75D}$, PurM$^{F105A}$, HisF$^{V126P}$). Cells were incubated for 6 h with carbenicillin at their respective 2× MIC (Control and LeuB$^{I134P}$: 25 μg/mL; purine mutants: 50 μg/mL) and colony-forming units (CFU) were determined on minimal agar medium after 48 h of incubation. CFU/mL are shown for two time points, before carbenicillin addition 0 h (black), and after 6 h (gray), (*n* = 3 distinct samples). Before treatment with carbenicillin all cultures reached exponential phase and optical densities (OD) between 0.2 and 0.8. (**C**) Time-kill assays with the control strain, and the HisF$^{V126P}$, PurM$^{F105A}$ and PurA$^{L75D}$ strains. Strains were incubated in minimal glucose medium and carbenicillin (50 μg/mL) for the time period indicated on the *x* axis (*n* = 3 replicates). Left: time-kill assays without adenine supplementation, right: time-kill assays with supplementation of 1 mM adenine. 50 μg/mL carbenicillin is 4× MIC of all strains in the presence of adenine. Before treatment with carbenicillin all cultures reached exponential phase and optical densities (OD) between 0.2 and 0.8. Source data are available online for this figure.

growing purine mutants (PurM$^{F105A}$, PurA$^{L75D}$ and HisF$^{V126P}$) were not resistant against gentamicin (Appendix Fig. S10), and vice versa, the slow-growing PtsI$^{I330P}$ mutant was not resistant to carbenicillin (Appendix Fig. S10).

Apart from antibiotic resistance, antibiotic tolerance has been associated with slow growth, especially in the case of β-lactams (Lee et al, 2018). Antibiotic tolerance has been defined as the ability to survive transient exposure to an antibiotic without changes in the MIC (Brauner et al, 2016). Therefore, we used the purine mutants to investigate whether their slow growth confers tolerance to carbenicillin or if the purine bottleneck has an additional survival benefit. The purine mutants, the LeuB$^{I134P}$ strain and the control strain were cultivated in minimal medium until they reached exponential phase and optical densities (OD) between 0.2 and 0.8. After a 6-hour treatment with carbenicillin, the purine mutants showed significantly higher survival compared to the control strain ($P$ value < 0.05), even when we adjusted the carbenicillin concentration such that each strain was treated at their respective 2× MIC (Fig. 3B; Appendix Fig. S11). However, the higher tolerance to carbenicillin of the purine mutants is likely due to their reduced growth rates, because the slow-growing mutant LeuB$^{I134P}$ showed a similar tolerance level, with an average survival rate of 11%, comparable to those of the purine mutants (PurA$^{L75D}$: 8%; PurM$^{F105A}$: 32%; HisF$^{V126P}$: 33%). Nevertheless, the tolerance phenotype was metabolism-dependent, because with the addition of adenine the kill curves of the control strain and the purine mutants converged (Fig. 3C). While adenine increased killing in the mutants, it decreased killing in the control strain, suggesting that adenine reduces tolerance of the mutants, but enhances survival of the control strain.

In summary, metabolism has pathway-specific effects on carbenicillin and gentamicin resistance, which are unrelated to growth rates. Tolerance against carbenicillin, however, is likely due to slow growth of the PurM$^{F105A}$, PurA$^{L75D}$ and HisF$^{V126P}$ mutants, as shown by the slow-growing LeuB$^{I134P}$ strain. This tolerance against carbenicillin is probably one factor that contributes to the evolution of higher resistance levels (Liu et al, 2020), as evidenced by the >10× MIC increases found in the PurA$^{L75D}$ strain after 72 h exposure to carbenicillin (Fig. EV4). However, multiple mechanisms may contribute to this evolvability, including higher tolerance and resistance, stress responses (Pribis et al, 2022) or changes in mutation rates (Windels et al, 2019).

## Clinical *E. coli* have metabolic bottlenecks, and a mutation in *purK* confers carbenicillin/sulbactam tolerance

To understand the clinical relevance of metabolic mutations, we examined 235 *E. coli* strains from different clinical isolates of the Tübingen University Hospital (Dataset EV4). To identify metabolic mutations in these isolates, we used metabolome analysis of strains with a growth defect in minimal glucose medium (Fig. 4A). Out of 235 strains, 41 strains showed a growth defect (Fig. EV5; Dataset EV4), and we measured their metabolome after a shift from rich medium to minimal glucose medium by flow-injection mass spectrometry (FI-MS) (Farke et al, 2023). FI-MS detected 636 ions with distinct masses that matched 811 metabolites. In 38 out of 41 isolates, at least one metabolite showed strong increases (fold change >8, Fig. 4B; Dataset EV5). Across these 38 isolates, 44 metabolites showed fold changes >8 and they mapped to 14

functional categories of the *E. coli* metabolic model iML1515 (Monk et al, 2017), which included amino acid biosynthesis and purine nucleotide biosynthesis (Dataset EV5). Next, we focused on three isolates EC-61, EC-96, and EC-244, which had high levels of cystathionine, histidinol-phosphate, and 5-amino-1-(5-phospho-β-D-ribosyl)imidazole (air), respectively. Cystathionine is involved in methionine biosynthesis and is metabolized by MetC, which had four resistance mutations in the CRISPR screen against carbenicillin (Fig. 1C). Histidinol-phosphate and air are intermediates in biosynthetic pathways of histidine and purine nucleotides, which were also targets for carbenicillin resistance mutations. We assumed that increases of these biosynthetic intermediates indicate bottlenecks in their respective biosynthesis pathways. To test this, we grew the three isolates in minimal glucose medium which we supplemented either with methionine, histidine, or adenine. Indeed, supplementing these metabolites markedly improved the growth of the respective isolate (Fig. 4C), demonstrating that pathway-specific metabolic bottlenecks occur in clinical *E. coli* isolates.

Next, we followed up on the purine bottleneck in EC-244, which was a urinary tract isolate. First, we used targeted metabolomics to measure the concentration of nucleotides and air in EC-244 and compared it to EC-249, which is another urinary tract isolate with normal growth. As expected, ATP, ADP, and AMP levels were lower in EC-244 compared to EC-249 (Appendix Fig. S12), and the LC-MS/MS data confirmed the high air levels detected by FI-MS (Fig. 4D). Notably, adenine feeding restored ATP levels and air levels in EC-244 (Fig. 4D; Appendix Fig. S12). Thus, the high levels of air, together with low purine end-products, suggested that EC-244 has a metabolic bottleneck in the middle of the purine pathway (Fig. 4E). To locate the bottleneck, we sequenced the genomes of EC-244 and EC-249 and compared sequences of the suspected purine genes to those of the laboratory *E. coli* strain BW25113. In total, we found 122 mutations in the 5 purine genes and 14 of them resulted in amino acid changes (Table EV2). However, only one amino acid change was unique to EC-244: PurK$^{E49G}$. PurK sequences of *E. coli* isolates in the NCBI pathogen database suggest that PurK$^{E49G}$ is a low frequency mutation, because none of the isolates in the database had this mutation. In contrast, the other 4 PurK mutations had higher frequencies, as they occurred in >364 isolates in the NCBI pathogen database (Appendix Fig. S13). Therefore, we assumed that PurK$^{E49G}$ is a hypomorphic allele that is responsible for the purine nucleotide synthesis bottleneck in EC-244. To test this hypothesis, we inserted the PurK$^{E49G}$ mutation into the laboratory *E. coli* strain BW25113, which indeed led to a purine auxotrophy (Fig. 4F), confirming its causative role in the purine nucleotide synthesis bottleneck of EC-244. Interestingly, the PurK$^{E49G}$ mutation led to a complete purine auxotrophy in *E. coli* BW25113, but not in EC-244. An explanation is that the other amino acid changes in PurK of EC-244 compensate the strong fitness defect caused by the PurK$^{E49G}$ mutation.

Based on the results with our CRISPR purine mutants, we expected that the purine limitation impaired the efficacy of carbenicillin in EC-244. However, EC-244 was highly resistant against carbenicillin, probably due to the expression of a β-lactamase (Dataset EV4). Therefore, we examined the killing activity of carbenicillin in the presence of the β-lactamase inhibitor sulbactam in EC-244. While carbenicillin/sulbactam exhibited no killing activity in EC-244 during a 6-h treatment, supplementation of adenine restored killing activity of carbenicillin/sulbactam such that only 0.01% of the adenine-fed cells survived the 6-h treatment

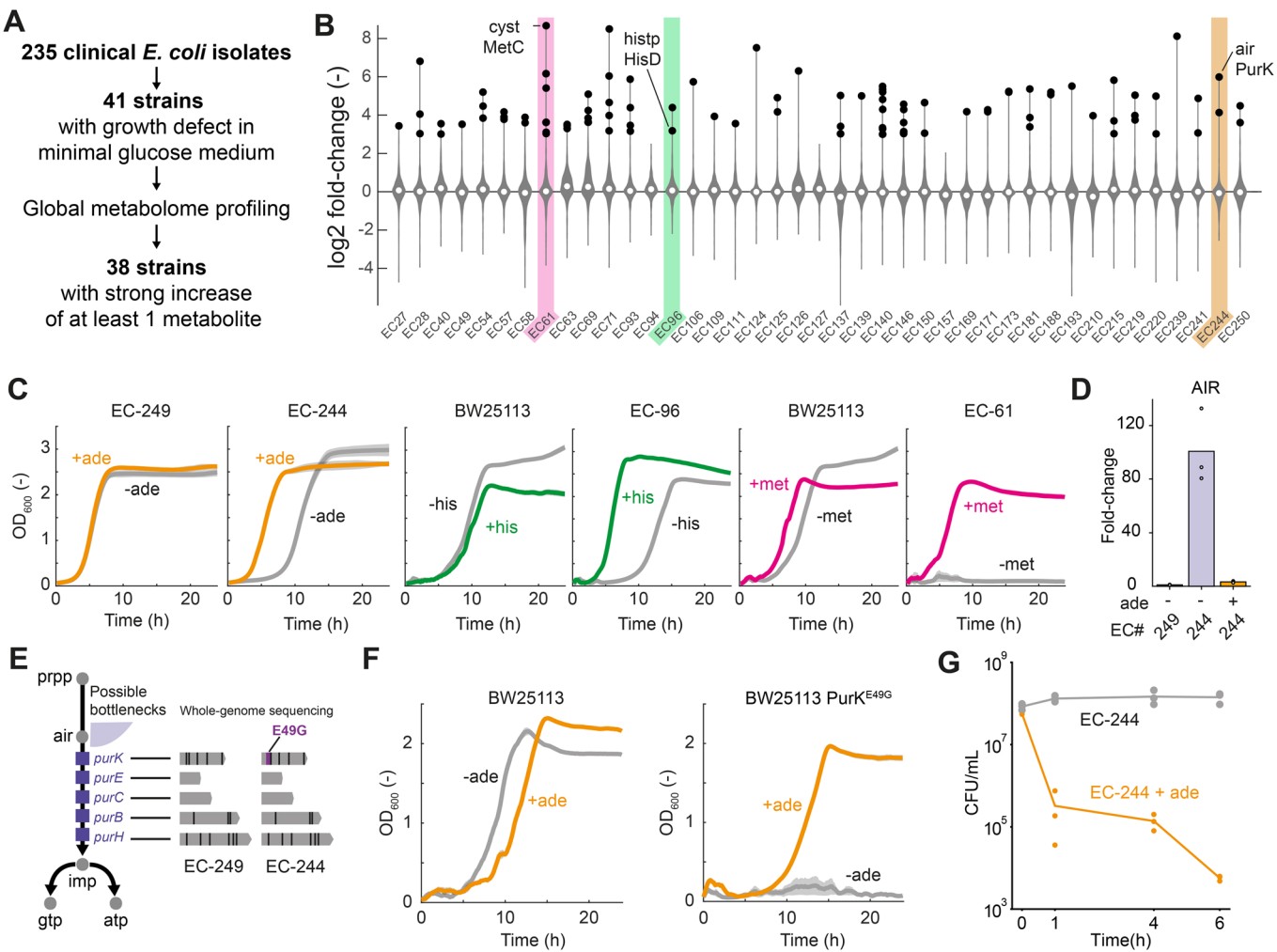

**Figure 4. Metabolome profiling identifies metabolic bottlenecks in clinical *E. coli* isolates and a purine bottleneck caused by PurK^E49G.**

(A) 235 *E. coli* isolates were obtained from different clinical specimens and cultivated in minimal glucose medium. 41 strains with growth defects (*n* = 2 cultures in 96-well plates) were selected for metabolome profiling with flow-injection time-of-flight mass spectrometry. (B) Metabolome profile of 41 clinical isolates (mean of *n* = 2 cultures in 96-well plates). Violin plot shows the distribution of 636 *m/z* features that were annotated to metabolites. *M/z* features with fold change >3 are shown in black. Three isolates (EC-61, EC-96, and EC-244) are annotated due to their accumulation of *m/z* features that were annotated to cystathionine (cyst), histidinol-phosphate (histp) and 5-amino-1-(5-phospho-β-D-ribosyl)imidazole (air). (C) Growth of EC-244 and EC-249 with (orange) and without (gray) supplementation of adenine. Shown are also growth curves of two clinical isolates EC-96 and EC-61 in minimal glucose medium supplemented with ʟ-histidine (green) or ʟ-methionine (magenta). Growth data of the *E. coli* wild-type BW25113 is shown as reference. Growth curves show means from *n* = 3 minimal glucose medium cultures grown in a plate reader. Gray areas show the standard deviation. (D) Relative concentrations of air in EC-244 (with and without adenine) and EC-249 (without adenine). Data are normalized to EC-249 (without adenine) and are represented as mean of *n* = 3 distinct samples (black dots). (E) Whole-genome sequencing was performed with EC-244 and EC-249. Sequences of genes of the purine de novo pathway were aligned to the *E. coli* wild-type BW25113 reference genome. (F) A mutation in *purK* (PurK^E49G) was identified in EC-244 and not in EC-249 and was inserted into the *E. coli* wild-type BW25113. Shown is the growth of the control strain and the PurK^E49G mutant in minimal glucose medium. Strains were grown either without adenine (gray), or with 1 mM of adenine (orange) (*n* = 3 distinct samples). (G) Time-kill assays with EC-244 without adenine (gray) and with supplementation of adenine (orange). EC-244 was incubated with 100 µg/mL carbenicillin and 12.5 µg/mL sulbactam for the time period indicated on the *x* axis (*n* = 3 replicates). Before treatment with carbenicillin/sulbactam both cultures reached exponential phase and optical densities (OD) between 0.2 and 0.8. Source data are available online for this figure.

(Fig. 4G). The purine limitation of EC-244 therefore leads to carbenicillin tolerance, but probably due to the slow growth rate without adenine.

## Discussion

In our study, we tested the resistance of 15,120 *E. coli* mutants, each with an amino acid change in an essential gene, against two

antibiotics of the β-lactam and aminoglycoside classes. Most mutations that led to antibiotic resistance were located in metabolic genes (95% on carbenicillin and 84% on gentamicin), although 31% of the mutations detected in our CRISPR screen were not in metabolic genes. We found that resistance from these metabolic mutations does not result from a global fitness defect, but local defects in specific pathways. For instance, 39% of the mutations that conferred resistance against carbenicillin were linked to genes involved in the purine nucleotide biosynthesis pathway. Similarly,

54% of the mutations conferring resistance to gentamicin were associated with the respiratory chain.

While our results show that resistance is due to bottlenecks in specific metabolic pathways, such as purine biosynthesis and respiratory chain-related processes, this does not exclude the possibility that global growth effects contribute to the resistance phenotype. However, our data suggest that global effects alone, such as reduced growth rates or reduced fitness, are insufficient to induce resistance phenotypes. The bottlenecks in specific metabolic pathways induce low-level antibiotic resistance: 2–4× MIC increases for carbenicillin and 2–8× MIC increases for gentamicin. This finding is consistent with previous reports about metabolic mutations that occurred during evolution of a laboratory *E. coli* strain (Lopatkin et al, 2021) and future studies must clarify if metabolic mutations are generally restricted to low-level resistance.

For gentamicin, the resistance mechanism likely involves the known dependency on oxidative metabolism for uptake of aminoglycosides (Taber et al, 1987). In the case of carbenicillin, several mechanisms have been proposed to explain why purine nucleotide biosynthesis influences antibiotic efficiency, for example, an antibiotic-induced adenine limitation that increases purine biosynthesis (Yang et al, 2019) or changes in ATP levels (Lopatkin et al, 2019; Conlon et al, 2016). However, because the PtsI$^{I330P}$ mutant had low ATP levels and was not resistant to carbenicillin, we assume that low ATP levels alone are not sufficient to confer resistance and that a bottleneck in de novo purine biosynthesis is required. Another hypothesis is that purine bottlenecks change carbenicillin transport, either because nucleotide metabolism interacts with membrane permeability via the porin OmpF (Zhao et al, 2021), or because of the higher efflux activities of auxotrophs (Yu et al, 2022). Our observation that purine bottlenecks lead to MIC increases and changes of several transporters supports the transport hypothesis. Although OmpF expression did not show changes at the transcriptional level (Dataset EV3), its regulation is known to occur post-transcriptionally through small regulatory RNAs.

We identified a purine metabolic bottleneck in a clinical *E. coli* isolate (EC-244), likely due to the E49G mutation in PurK. We cannot conclude that the acquisition of the PurK$^{E49G}$ mutation is the result of β-lactam selective pressure. Instead, it is more likely that PurK$^{E49G}$ was acquired in an environment where de novo purine synthesis is not essential. For example, human urine has been shown to complement deletions of genes in purine biosynthesis (Ma et al, 2018), and *E. coli* can salvage nucleotides present in urine (Andersen-Civil et al, 2018). Nevertheless, the de novo purine pathway has been shown to be essential for survival and colonization in niches such as the gut (Vogel-Scheel et al, 2010), human blood (Samant et al, 2008), or inside host cells (Shaffer et al, 2017). The PurK$^{E49G}$ mutation was not included in the NCBI pathogen database, indicating that the mutation is rare. Therefore, detecting mutations in metabolic genes of clinical or environmental genomes does not necessarily indicate a metabolic bottleneck or potential resistance or tolerance. Future research could investigate the prevalence of metabolic mutations in bacterial genomes and assess whether these mutations induce a bottleneck and if they affect antibiotic action.

In conclusion, our results demonstrate that bacterial metabolism plays an important role for antibiotic resistance and tolerance, which in turn implies that the nutritional environment at an infection site is equally important. We used minimal media to control the supply of metabolites and to systematically evaluate the response of *E. coli* to the availability of nutrients like adenine. This is not feasible with complex media like Mueller-Hinton broth, which contains variable components that may deplete unevenly during an experiment. Although our experimental conditions differ from in vivo environments, our findings contribute to a more detailed understanding of how metabolic mutations can simultaneously influence both resistance and tolerance mechanisms. Given that the effect of metabolic mutations is highly condition- and nutrient-dependent, our results emphasize the need for new approaches to treat bacterial infections, either by considering metabolic strain level variation or by targeting the extracellular environment at an infection site to maximize the efficacy of an antibiotic.

## Methods

### Reagents and tools table

| Reagent/resource | Reference or source | Identifier or catalog number |
|---|---|---|
| **Experimental models** | | |
| *E. coli* BW25113 | Baba et al, 2006 | |
| YYdCas9: BW25993 *intC::tetR-dcas9-aadA lacY::ypet-cat* | Lawson et al, 2017 | N/A |
| *E. coli* BW25113 // pTS041 // pTS040 | Schramm et al, 2023 | N/A |
| *E. coli* BW25113// pTS041 // pTS040 // purA$^{L75D}$ | This study | N/A |
| *E. coli* BW25113// pTS041 // pTS040 // purK$^{E49G}$ | This study | N/A |
| *E. coli* BW25113// pTS041 // pTS040 // purM$^{F105A}$ | This study | N/A |
| *E. coli* BW25113// pTS041 // pTS040 // hisF$^{V126}$ | This study | N/A |
| *E. coli* BW25113// pTS041 // pTS040 // ptsI$^{I330P}$ | This study | N/A |
| *E. coli* BW25113// pTS041 // pTS040 // hemA$^{L276Q}$ | This study | N/A |
| *E. coli* BW25113// pTS041 // pTS040 // ispE$^{V146W}$ | This study | N/A |
| *E. coli* BW25113// pTS041 // pTS040 // pgsA$^{V44P}$ | This study | N/A |
| *E. coli* BW25113// pTS041 // pTS040 // ribD$^{L384W}$ | This study | N/A |
| *E. coli* BW25113// pTS041 // pTS040 // leuB$^{I134P}$ | This study | N/A |
| **Recombinant DNA** | | |
| pTS40 | Schramm et al, 2023 | N/A |
| pTS41 | Schramm et al, 2023 | N/A |
| pgRNA-bacteria | Qi et al, 2013 | Addgene #44251 |
| **Oligonucleotides and other sequence-based reagents** | | |
| PCR primer | This study | Table EV3 |

| Reagent/resource | Reference or source | Identifier or catalog number |
|---|---|---|
| **Chemicals, enzymes, and other reagents** | | |
| Ammonium carbonate | Honeywell | Cat#10361-29-2 |
| Isopropanol, Rotisolv ≥99.95%, Ultra LC-MS | Roth | Cat#0733.1 |
| Acetonitrile | Honeywell | # 14261-1l |
| Methanol | VWR | # 83638.320 |
| HP-921 and purine in API-TOF Reference Mass Solution Kit | Agilent | G1969-85001 |
| Gentamicin | Roth | #O233.2 |
| Carbenicillin | Roth | #6344.3 |
| Adenine | Sigma | #A8751-1G |
| 5-aminolevulinic acid | Merck | # A7793 |
| L-histidine | Merck | # H8000 |
| L-methionine | Merck | # 64319 |
| glucose-6-phosphate | | Merck #G7879 |
| anhydrotetracycline | Cayman Chemicals | #10009542 |
| DNase I | Roche | #09852093103 |
| **Software** | | |
| Matlab R2021a | mathworks.com | N/A |
| MSConvert | https://proteowizard.sourceforge.io | N/A |
| nf-core/rnaseq v3.16.0 | https://doi.org/10.5281/zenodo.1400710 | |
| **Other** | | |
| Illumina NextSeq 500 | Illumina | – |
| Agilent 6495 qTOF mass spectrometer | Agilent Technologies | – |
| Agilent 6495 triple quadrupole mass spectrometer | Agilent Technologies | – |
| BioTek Synergy plate reader | BioTek | – |
| Qubit RNA BR Assay Kit | Thermo Fisher | |
| RNA 6000 Pico kit | Agilent | #5067-1513 |

## Methods and protocols

### Strains

*E. coli* BW25113 (Baba et al, 2006) was used to construct the CRISPR library and the strains $PurA^{L75D}$, $PurM^{F105A}$, $HisF^{V126P}$, and $PurK^{E49G}$. *E. coli* YYdCas9 (Lawson et al, 2017) was used for CRISPRi. One Shot™ TOP10 *E. coli* (Thermo Fischer #C404010) was used for intermediate cloning. Clinical isolates were obtained from the Institute for Medical Microbiology and Hygiene (Tübingen) and were validated using MALDI-TOF mass spectrometry.

### Media

Cultivations were performed in LB medium (Sigma #L3522) or M9 minimal medium with glucose as sole carbon source (5 g/L). M9 medium was composed by (per liter): 7.52 g $Na_2HPO_4$ 2 $H_2O$, 5 g $KH_2PO_4$, 1.5 g $(NH_4)_2SO_4$, 0.5 g NaCl. The following components were sterilized separately and then added (per liter of final medium): 1 mL 0.1 M $CaCl_2$, 1 mL 1 M $MgSO_4$, 0.6 mL 0.1 M $FeCl_3$, 2 mL 1.4 mM thiamine-HCl and 10 mL trace salts solution. The trace salts solution contained (per liter): 180 mg $ZnSO_4$ 7 $H_2O$, 120 mg $CuCl_2$ 2 $H_2O$, 120 mg $MnSO_4$ $H_2O$, 180 mg $CoCl_2$ 6 $H_2O$. When needed, M9 agar plates were done by mixing (1:1) a 2× M9 solution with 30 g/L molten agar (Roth #5210.2). Kanamycin (50 μg/mL; Roth #T832.3) and chloramphenicol (30 μg/mL; Merck #C0378-25G) were added in either liquid or agar medium when strains harbored pTS040 (chloramphenicol resistance marker) and pTS041 (kanamycin resistance marker). When needed, gentamicin (Roth #O233.2) or carbenicillin (Roth #6344.3) were added to the M9 medium at various concentrations specified for each experiment. Adenine (Sigma #A8751-1G) was added to agar and liquid medium at a final concentration of 1 mM, and to agar and liquid medium at final concentration 100 μM for experiments with hospital bacterial isolates. 5-aminolevulinic acid (Merck # A7793) was added to agar and liquid medium at a final concentration of 10 mg/mL. L-histidine (Merck # H8000) and L-methionine (Merck # 64319) were added to liquid medium at 200 μM. When required, glucose-6-phosphate (Merck #G7879) was added to agar and liquid minimal mediums in replacement of glucose at a final concentration of 1 g/L. To induce dCas9 expression, the CRISPRi experiments were performed with 0.2 μM of anhydrotetracycline (aTc; Cayman Chemicals #10009542) in liquid medium and 1 μM aTc in agar medium.

### Screening of antibiotic resistance of the CRISPR library

Antibiotic resistance was screened in a pooled CRISPR library with 15,120 *E. coli* mutants that were constructed previously (Schramm et al, 2023). The CRISPR library and the control strain were each cultivated in 10 mL LB medium at 30 °C until $OD_{600}$ reached 0.5. An equivalent of $OD_{600} = 5$ was then centrifuged at 30 °C and pellets were resuspended in 10 mL of fresh M9 medium. Centrifugation was repeated to remove traces of LB medium. Cells were resuspended in 10 mL of M9 medium and further incubated at 30 °C for 1 h. $OD_{600}$ was then set to 0.5 and 500 μL were used to inoculate $150 \times 20$ mm M9 agar plates. Plates were then incubated at 37 °C for 48 h. After incubation, plates were imaged with an Epson V370 scanner. If necessary, colonies were counted. Then, colonies were harvested from each plate using 7.5 mL of LB medium. $OD_{600}$ was measured and an equivalent of $OD_{600} = 10$ was pelleted in a microcentrifuge tube. Plasmids were then purified by miniprep (Thermo scientist, GeneJET Plasmid Miniprep Kit) for amplification of repair templates and sgRNA (barcodes). Hereafter, 3 ng of plasmid DNA was used for amplification (15 cycles) of the barcodes using two primers suited for further indexing PCRs:

forward primer:

5'-TCGTCGGCAGCGTCAGATGTGTATAAGAGACAGG-TATCACGAGGCAGATCCTCTG-3'

reverse primer:

5'-GTCTCGTGGGCTCGGAGATGTGTATAAGAGACA-GACTCGGTGCCACTTTTTCAAGTT-3'

Amplicons were purified by AMPure XP PCR beads (Beckman Coulter, #A63881). Using standard Illumina indexing primers, amplicons were indexed in a second PCR and again purified by bead-clean up. Amplicons were pooled and sequenced on an Illumina NextSeq 500 (paired-end, NextSeq™ 500 Mid Output Kit v2.5, #20024908, 300 cycles).

### Illumina sequencing data analysis

Demultiplexed paired-end reads were aligned, merged (based on overlapping sequences), and trimmed to the region of interest using a custom Matlab script. The resulting processed reads were mapped against the designed sequences of the library. For each library member, the number of matching reads was counted. Only reads that shared a 100% identity with a designed sequence were considered for further analysis. Read counts of 0 were set to 1, to avoid division by 0. Read counts lower than 15 on both reference plates and antibiotic plates were not considered in the analysis. Read counts were normalized by dividing the read counts of each mutants by the total number of reads in a given sample. Fold changes were calculated by dividing normalized read counts of each mutant on the antibiotic plates by normalized read counts on the reference plates obtained from the same experiment.

### Construction of single CRISPR strains

The $PurK^{E49G}$, $PurA^{L75D}$, $PurM^{F105A}$, and $HisF^{V126P}$ strains were reconstructed with the same method as the CRISPR library. Plasmid pT0S41 was first transformed with electroporation into WT BW25113. Plasmids pTS040 were built by assembling the pTS040 backbone with oligonucleotides (Twist Bioscience) encoding sgRNA and homology arms associated with the desired mutations. Then, plasmids pTS040 were transformed with electroporation after 30 min induction with 7.5 g/L arabinose (lambda red expression). Strains were cultivated for 1 h in SOC medium with kanamycin and 1 μM aTc to induce Cas9 expression. Strains were then plated on LB agar with kanamycin, chloramphenicol and 1 μM aTc. Incubation was done at 37 °C overnight. Subsequently, single colonies were picked for colony PCR to amplify the potentially mutated genes of interest. PCR amplicons were purified (Macherey-Nagel #740609) and used for sequencing (Eurofins genomics). Sequences were analyzed with the Benchling software and the MAFFT algorithm. Strains with the correct mutations were cultivated overnight in 4 mL LB with kanamycin and chloramphenicol to prepare glycerol stocks. Mutants $PtsI^{I330P}$, $RibD^{L364W}$, $PgsA^{V44P}$, $IspE^{V146W}$, and $HemA^{L276Q}$ were isolated directly from the CRISPR library after gentamicin challenge and mutations were confirmed by sequencing the genomic region as described above. The cloning and isolation of the $LeuB^{I134P}$ mutant has been described previously (Schramm et al, 2023).

### Construction of CRISPRi strains

Plasmids pgRNAK-purA#1 (protospacer: 5'-TTTACCTTCGT-CACCCCATT-3') and pgRNAK-mRFP (protospacer: 5'-AACTTT-CAGTTTAGCGGTCT-3') were constructed by exchanging the ampicillin resistance cassette of the plasmid pgRNA (Addgene #44251)(Qi et al, 2013) with a kanamycin resistance cassette. Plasmids were then transformed into the E. coli strain $YYdCas9^{37}$ using electroporation.

### Generation of growth curves and determination of growth rates

Precultures in 4 mL LB were inoculated from glycerol stocks for 8 h at 37 °C 220 RPM shaking and transferred to M9 medium for overnight incubation at 37 °C and 220 RPM. M9 precultures in the exponential phase were used to inoculate 96-well plates at starting $OD_{600} < 0.05$. Incubation was performed for 24 h at 37 °C. Various plate readers were used (BioTek Epoch, BioTek Synergy, Tecan Infinite 200 Pro, Tecan Spark). Therefore, $OD_{600}$ values were corrected based on former calibration experiments

For generation of adenine feeding growth data (for the BW25113 $PurK^{E49G}$ and control strain), precultures in 4 mL LB were inoculated from glycerol stocks for 6 h at 37 °C 220 RPM shaking. Then, 2 mL of each culture was centrifuged at maximum speed for 5 min. The supernatant was discarded, and pellet was resuspended in 2 mL of M9 medium with glucose. Washing and resuspension was repeated. $OD_{600}$ was then measured and normalized across the strains. A 96-well plate with M9 medium supplemented with or without 1 mM adenine was then inoculated with the washed cultures. $OD_{600}$ was measured in a BioTek Synergy plate reader for 24 h at 37 °C.

Growth rates were determined with the following method: coefficients of determination ($R^2$) were calculated over a 2 h time window. Only arrays with $R^2 > 0.99$ were selected. Growth rates were determined by linear regression and maximal growth rates were selected. For clinical isolates Area Under Curve (AUC) determination, six 96-well deep-well plates were prepared with 500 μL of LB medium without antibiotics per well. Strains EC-1 to EC-256 were added each to a single well from glycerol stocks in duplicates. Plates were sealed with Breathe-Easy foils and incubated for 6 h at 37 °C 220 RPM. Then, 50 μL of each well was used to inoculate any of six deep-well plates with 450 μL M9 medium with glucose and no antibiotics (1:10 dilution). This operation was repeated for a final 1:100 dilution of the LB inoculum in M9 medium. Six 96-well plates were prepared with each 135 μL of M9 medium with glucose and no antibiotics. 15 μL in wells of the previously prepared deep-well plates were used to inoculate these 96-well plates. The plates were sealed with a lid and parafilm and incubated for 24 h at 37 °C in BioTek Logphase 600 plate readers. The AUC was determined as the integral of ODs between 0 and 12 h via the trapezoidal method with unit spacing (trapz.m MATLAB function).

### Agar dilution assays to measure MICs

M9 agar plates with various concentrations of antibiotics and additives were prepared as described above. Precultures in 4 mL LB were inoculated from glycerol stocks for 8 h at 37 °C and 220 rotations per minutes (RPM) shaking and transferred to M9 medium for overnight incubation at 37 °C and 220 RPM. Before starting the assay, precultures were reinoculated and grown in fresh M9 medium to obtain exponentially growing cells. Then, cultures were diluted with fresh M9 medium to set $OD_{600} = 0.1$. A 96-well plate was prepared with three wells containing 135 μL of fresh M9 medium for each strain to be spotted. Each preculture was then 1:10 diluted by mixing 15 μL of the 0.1 $OD_{600}$ preculture with 135 μL of fresh medium. This process was repeated two times to generate the 1:100 and 1:1000 dilutions. 7 μL of each dilution was then added to the agar plate with a multi-channel pipette to generate the spots. Spots were then left to dry under a flame and plates were incubated for 48 h at 37 °C. After incubation, plates were imaged with an Epson V370 scanner. MIC was determined based on the first concentration at which a full spot of cells does not grow at $OD_{600}$ 0.0001 (Wiegand et al, 2008).

### Time-kill assay

Precultures in 4 mL LB were inoculated from glycerol stocks for 8 h at 37 °C and 220 RPM and transferred to M9 medium for overnight

incubation at 37 °C and 220 RPM. M9 precultures were used to re-inoculate shake flasks containing 25 mL of M9 medium with kanamycin and chloramphenicol. When $OD_{600}$ 0.25 was reached, 10 mL of medium were transferred to a new shake flask and carbenicillin was added at a final concentration 50 µg/mL or 25 µg/mL depending on the strain tested. Cells were incubated at 37 °C and 220 RPM. At each time point, 1 mL of each culture was sampled into a microcentrifuge tube and cells were centrifuged at 3000 RPM for 10 min. The supernatant was discarded, and cells were resuspended in 1 mL of fresh M9 medium without carbenicillin. This washing step was then repeated. Cells were then serial diluted in fresh M9 medium by a factor of 10 to obtain dilutions of 1:100, 1:1000, 1:10,000 and 1:100,000. In total, 100 µL from each dilution were then plated on M9 agar and incubated 48 h at 37 °C. Colonies were then counted to quantify colony-forming units (CFUs) per mL.

For time-kill assays with hospital bacterial isolates, precultures of EC-244 were first performed in LB for 8 h at 37 °C. Each preculture was then split into one M9 preculture with 100 µM adenine and one preculture without adenine, and grown overnight at 37 °C. The next day, cultures in the exponential phase were used to inoculate shake flasks with 25 mL of M9 medium with or without 100 µM adenine, with starting $OD_{600}$ 0.25. Sulbactam (TCI #S0868) was added at a final concentration of 12.5 µg/mL and carbenicillin was added at final concentration of 100 µg/mL. At indicated time points, 1 mL of culture were sampled from each flask and cells were washed with M9 medium supplemented with 5 g/L glucose and 100 µM adenine and plated on M9 agar supplemented with 100 µM adenine.

### Evolution of higher carbenicillin resistance

Precultures in 4 mL LB were inoculated from glycerol stocks for 8 h at 37 °C 220 RPM shaking and transferred to M9 medium for overnight incubation at 37 °C and 220 RPM. The $OD_{600}$ of the M9 precultures in the exponential phase were normalized to 0.01 and 100 µl of cultures were used to inoculate M9 agar medium plates with either 0, 3.1 or 6.2 µg/ml. Concentration of 3.1 µg/ml were used for the incubation of the control strain and concentrations of 6.2 µg/ml were used for the mutant strains. Plates were incubated 72 h at 37 °C. Colonies, corresponding to evolved strains, were then counted and picked for cultivation in LB medium and glycerol storage. Carbenicillin resistance of the evolved strains was then re-tested using agar dilution assay.

### Cultivation conditions for metabolome sampling

Precultures in 4 mL LB were inoculated from glycerol stocks for 8 h at 37 °C 220 RPM shaking and transferred to M9 medium for overnight incubation at 37 °C and 220 RPM. M9 precultures in the exponential phase were used to inoculate shake flasks containing 10 mL of M9 medium with a starting $OD_{600}$ of 0.1. Strains were cultivated in triplicates until $OD_{600}$ reached 0.25–0.6. For metabolomics, flasks were then rapidly transferred to a thermo-statically controlled hood at 37 °C and an equivalent of $OD_{600} = 1$ was sampled. For metabolome profiling of clinical isolates, 96-well deep-well plates were prepared with 1 mL of LB medium. The selected 41 strains with growth defects were added to the prepared plates from glycerol stocks in duplicate. The plate was sealed with Breathe-Easy foils (Diverse Biotech #BEM-1) and incubated for 6 h at 37 °C 220 RPM. The plate was then centrifugated at maximum

speed and 37 °C. The supernatant was discarded and pellets were each resuspended with 1 mL of M9 medium with glucose. The washing step was repeated. The plate was then incubated for 1 h 30 min at 37 °C and 220 RPM. Sampling was then performed by pelleting the cells in deep-well plates as described below.

### Metabolomics measurements

Cultivations were performed as described above. For targeted metabolomics, culture aliquots were vacuum-filtered on a 0.45-µm pore size filter (Merck Millipore #HVLP02500). Filters were immediately transferred into a 40:40:20 (v-%) acetonitrile (Honey-well # 14261-1 l)/methanol (VWR # 83638.320)/water extraction solution at −20 °C. Filters were incubated in the extraction solution for at least 30 min. Subsequently, metabolite extracts were centrifuged for 15 min at 13,000 RPM at −9 °C and the super-natants were stored at −80 °C until analysis. Metabolite extracts were mixed with a $^{13}C$-labeled internal standard in a 1:1 ratio. LC-MS/MS analysis was performed with an Agilent 6495 triple quadrupole mass spectrometer (Agilent Technologies) as described previously (Guder et al, 2017). An Agilent 1290 Infinity II UHPLC system (Agilent Technologies) was used for liquid chromatography. The temperature of the column oven was 30 °C, and the injection volume was 3 µL. LC solvents in channel A were either water with 10 mM ammonium formate and 0.1% formic acid (v/v) (for acidic conditions), or water with 10 mM ammonium carbonate and 0.2% ammonium hydroxide (for basic conditions). LC solvents in channel B were either acetonitrile with 0.1% formic acid (v/v) (for acidic conditions) or acetonitrile without additive (for basic conditions). LC columns were an Acquity BEH Amide (30 × 2.1 mm, 1.7 µm) for acidic conditions, and an iHILIC-Fusion(P) (50 × 2.1 mm, 5 µm) for basic conditions. The gradient for basic and acidic conditions was: 0 min 90% B; 1.3 min 40% B; 1.5 min 40% B; 1.7 min 90% B; 2 min 90% B. The ratio of $^{12}C$ and $^{13}C$ peak heights was used to quantify metabolites.

For flow-injection metabolomics, metabolite extracts were obtained by pelleting the cells in deep-well plates at maximum speed for 2 min at 37 °C. The supernatant was then discarded, and pellets were resuspended with 200 µL of 40:40:20 (v-%) acetonitrile/methanol/water extraction solution at −20 °C. The plates were sealed and incubated overnight at −20 °C. Pellets were then resuspended and centrifugated at maximum speed for 5 min at −9 °C. In all, 150 µL of supernatant from each well was transferred to a 96-well plate and stored at −80 °C for measurements. Extracts were directly injected into an Agilent 6546 Series quadrupole time-of-flight mass spectrometer (Agilent Technologies, USA) as described previously. The electrospray source was operated in negative and positive ionization mode. The mobile phase was 60:40 isopropanol:water buffered with 10 mM ammonium carbonate (NH4)2CO3 and 0.04% (v/v) ammonium hydroxide for both ionization modes, and the flow rate was 0.15 mL/min. For online mass axis correction, 2-propanol (in the mobile phase) and HP-921 were used for negative mode and purine and HP-921 were used for positive mode. Mass spectra were recorded in profile mode from 50 to 1700 $m/z$ with a frequency of 1.4 spectra/s for 0.5 min using 10 Ghz resolving power. Raw data files were converted into mzXML files and processed by custom MATLAB scripts. The 32 spectra with the highest signal in the total ion count were summed and baseline adjusted with *msbackadj.m*. Peaks with a minimum peak height of 5000 units and a peak prominence of 5000 units were

selected with *findpeaks.m*, and annotated with a 3 mDa tolerance by matching monoisotopic masses of all metabolites in the *i*ML1515 model (Monk et al, 2017), considering a single proton loss ([M-H]$^-$) in negative mode and single proton gain ([M + H]$^+$) in positive mode. Positive and negative mode annotation were merged and if a metabolite was annotated in both modes positive mode was selected. For each metabolite, the height of the annotated ion peak was taken for further analysis and normalized to the mean across the 41 isolates to obtain fold-change values.

### Transcriptomics for PurA$^{L75D}$ mutant and control strain

The PurA$^{L75D}$ mutant strain and the control strain were grown in minimal medium until they reached an OD$_{600}$ between 0.4 and 0.5. One OD$_{600}$-unit was pelleted by centrifugation, frozen, and RNA was extracted with the Quick-RNA Fungal/Bacterial Miniprep Kit from Zymo Biomics. Isolated RNA was treated with DNase I (Roche #09852093103). Total RNA was quantified with the Qubit RNA BR Assay Kit (Thermo Fisher), and RNA integrity was determined with an Agilent 2100 BioAnalyzer and the RNA 6000 Pico kit (Agilent #5067-1513). The library preparation was performed with Illumina Stranded Total RNA Prep, the ligation with the Ribo-Zero Plus Microbiome rRNA Depletion Kit (Illumina). Briefly, 100 ng of total RNA per sample were subjected to rRNA depletion, followed by cDNA library construction, adapter ligation, and 15 cycles of barcoding PCR. Generated libraries were quantified with Qubit 1x DNA HS Assay Kit (Thermo Fisher) and the fragment distribution was checked with Agilent 2100 BioAnalyzer using High Sensitivity DNA Kit (Agilent). Subsequently, libraries were pooled and sequenced on an Illumina NovaSeq 6000 device, using NovaSeq 6000 SP Reagent Kit v1.5 (200 cycles) with a run mode 100,10,10,100.

Data were processed using nf-core/rnaseq v3.16.0 (https://doi.org/10.5281/zenodo.1400710) of the nf-core collection of workflows (Ewels et al, 2020), utilizing reproducible software environments from the Bioconda (Grüning et al, 2018) and Biocontainers (da Veiga Leprevost et al, 2017) projects. The pipeline was executed with Nextflow v24.04.4 (Di Tommaso et al, 2017), a workflow management system. In the pipeline, cutadapt (v3.4) was used for adapter trimming. Afterward, hisat2 v2.2.1(Kim et al, 2019) was used for the mapping to the reference and feature counts from the R-package subread v2.0.1 (Liao et al, 2019) to get their abundance (R version 4.0.3). The feature count data were analyzed with a custom R script, which used DESeq2 v1.44.0 (Love et al, 2014) to get the fold changes between mutant and control. The genome of *E. coli* BW25113 was used as reference.

### Whole-genome sequencing of *E. coli* clinical isolates EC-244 and EC-249

DNA was extracted from EC-244 and EC-249 using the DNeasy UltraClean Microbial Kit (Qiagen), followed by library preparation (Illumina DNA Prep, (M) Tagmentation, Illumina) and barcoding (IDT for Illumina DNA/RNA UD Indexes). Sequencing was performed using a Mid Output Cartridge (NextSeq 500/550 Mid Output Kit v2.5 (300 Cycles)) on an Illumina NextSeq 500 machine. Following the sequencing, an alignment of the resulting fastq files was performed using bowtie2. The genome of *E. coli* BW25113 was used as reference. Subsequently, the alignment was investigated using the Integrative Genomics Viewer (Version 2.16.0). A comparison of mutations within the *purK* gene was performed

with genomes of clinical isolates from the NCBI Pathogens database (Sayers et al, 2022). A total of 9.369 genomes were collected, and PurK protein sequences were obtained for 4.352 genomes. Amino acid changes in these PurK protein sequences were identified by alignment with MUSCLE (Edgar, 2004).

### Statistical testing

All *P* values obtained in this manuscript were obtained by performing paired two-tailed *t* tests using custom python scripts (bioinfokit package) and Microsoft Excel. Volcano plots were made using custom python scripts (bioinfokit package). Pearson correlation coefficients (PCC) and coefficients of determination ($R^2$) were calculated with custom python scripts (sci-kit learn and SciPy packages).

## Data availability

Illumina sequencing data and transcriptome data are provided on the EMBL-EBI European Nucleotide Archive (ENA) online repository: ERP159795 (https://www.ebi.ac.uk/ena/browser/view/PRJEB75208). Metabolomics data are provided on the MassIVE repository: MassIVE MSV000094698 for targeted metabolomics (https://massive.ucsd.edu/ProteoSAFe/dataset.jsp?task=dbf28826aebd4272a0743bf4ce3ae70a) and MassIVE MSV000094699 for untargeted metabolomics (https://massive.ucsd.edu/ProteoSAFe/dataset.jsp?task=3a3e076019b24a0ca20122bdae93c514).

The source data of this paper are collected in the following database record: biostudies:S-SCDT-10_1038-S44320-024-00084-z.

## Peer review information

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

## Acknowledgements

The authors thank Libera Lo Presti, Urs Jenal, Heike Brötz-Oesterhelt and Ilka Bischofs-Pfeifer for discussions. This work was funded by the DFG Cluster of Excellence EXC2124 'Controlling Microbes to Fight Infection' (CMFI). Amplicon sequencing was supported by the Quantitative Biology Center (QBiC), Institute for Medical Genetics and Applied Genomics (IMGAG), and Institute for Medical Microbiology and Hygiene (MGM) of the University of Tübingen. NGS sequencing methods were performed with the support of the DFG-funded NGS Competence Center Tübingen (INST 37/1049-1) and the Institute for Molecular Microbiology and Hygiene, University Clinic Tübingen. Data management and storage of raw data for this project were supported by the Quantitative Biology Center (QBiC), University of Tübingen, Germany. We acknowledge support from the Open Access Publication Fund of the University of Tübingen.

## Author contributions

**Paul Lubrano**: Conceptualization; Data curation; Formal analysis; Investigation; Visualization; Methodology; Writing—original draft; Writing—review and editing. **Fabian Smollich**: Data curation; Formal analysis; Investigation; Visualization; Methodology; Writing—review and editing; Co-first author. **Thorben Schramm**: Investigation; Methodology. **Elisabeth Lorenz**: Investigation; Methodology. **Alejandra Alvarado**: Investigation; Methodology. **Seraina Carmen Eigenmann**: Investigation; Methodology. **Amelie Stadelmann**: Investigation; Methodology. **Sevvalli Thavapalan**: Investigation; Methodology. **Nils Waffenschmidt**: Investigation; Methodology. **Timo Glatter**: Investigation; Methodology. **Nadine Hoffmann**: Investigation; Methodology. **Jennifer Müller**: Investigation; Methodology. **Silke Peter**: Investigation; Methodology. **Knut Drescher**: Investigation; Methodology. **Hannes Link**: Conceptualization; Data curation; Formal analysis; Supervision; Funding acquisition; Validation; Investigation; Visualization; Methodology; Writing—original draft; Project administration; Writing—review and editing.

Source data underlying figure panels in this paper may have individual authorship assigned. Where available, figure panel/source data authorship is listed in the following database record: biostudies:S-SCDT-10_1038-S44320-024-00084-z.

## Funding

## Disclosure and competing interests statement

The authors declare no competing interests.

# Expanded View Figures

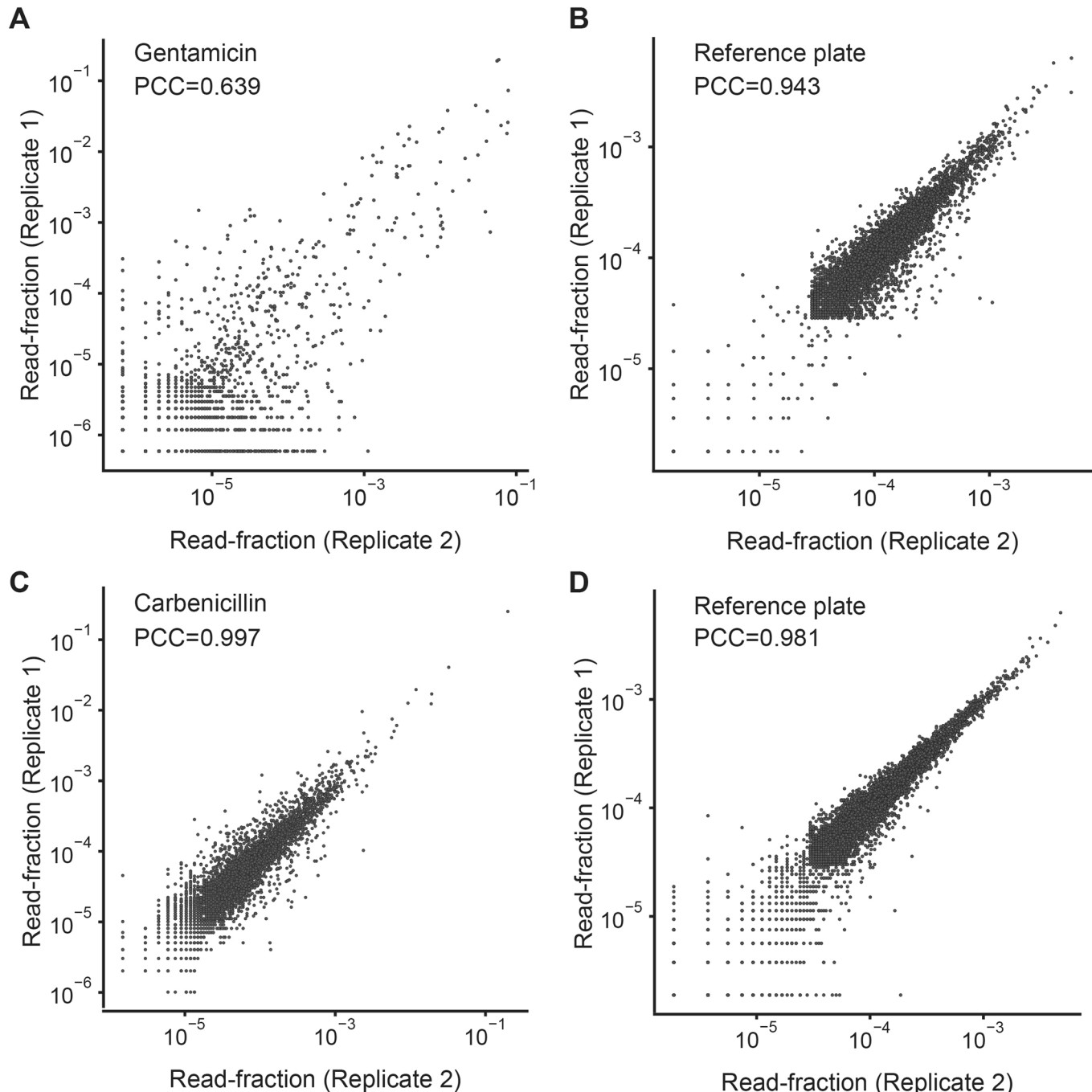

**Figure EV1. Read fractions of barcodes of mutants in the CRISPR library.**

(**A**, **B**) are barcode read fractions from the gentamicin screen. (**C**, **D**) are barcode read fractions from the carbenicillin screen. The Pearson correlation coefficient (PCC) is shown for the two replicates.

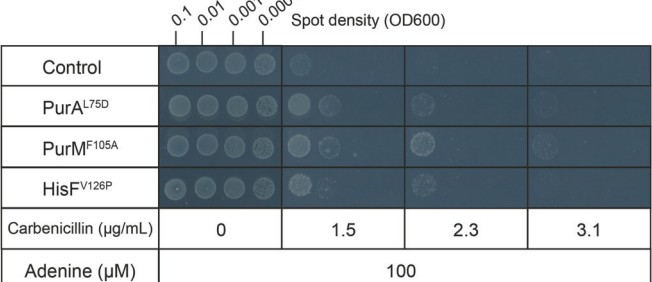

| | 0.1 0.01 0.001 0.0001 | Spot density (OD600) | | |
|---|---|---|---|---|
| Control | | | | |
| PurA^L75D | | | | |
| PurM^F105A | | | | |
| HisF^V126P | | | | |
| Carbenicillin (μg/mL) | 0 | 1.5 | 2.3 | 3.1 |
| Adenine (μM) | 100 | | | |

**Figure EV2.   Agar dilution assay with the control strain and three purine mutants (HisF^V126P, PurM^F105A and PurA^L75D).**

Each strain was spotted on agar plates with minimal glucose medium supplemented with 100 μM adenine and increasing concentrations of carbenicillin (MIC = 1.5 μg/mL). Plates were incubated 48 h. Shown is one of $n = 2$ replicates. Spot assays were performed on the same plate per concentration, and scans of plates with different concentrations were assembled into a single figure using Adobe Illustrator. Source data are available online for this figure.

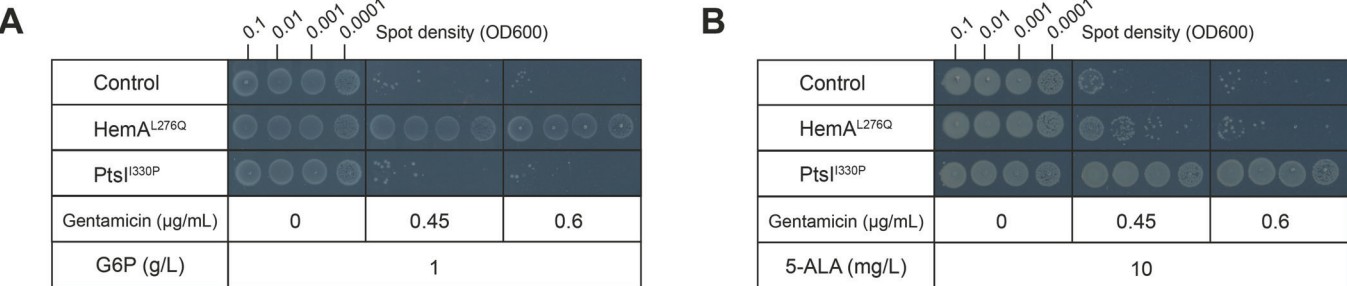

**Figure EV3.  Agar dilution assay with the control strain, the HemA[L276Q] strain and the PtsI[I330P] strain.**

(A) Each strain was spotted on agar plates with minimal medium containing glucose-6-phosphate (G6P) instead of glucose as carbon source, and with increasing concentrations of gentamicin (MIC = 0.45 µg/mL). (B) same as (A) but with glucose as carbon source and supplementation of 5-aminolevulinic acid (5-ALA). Plates were incubated 48 h. Spot assays were performed on the same plate per concentration, and scans of plates with different concentrations were assembled into a single figure using Adobe Illustrator. Source data are available online for this figure.

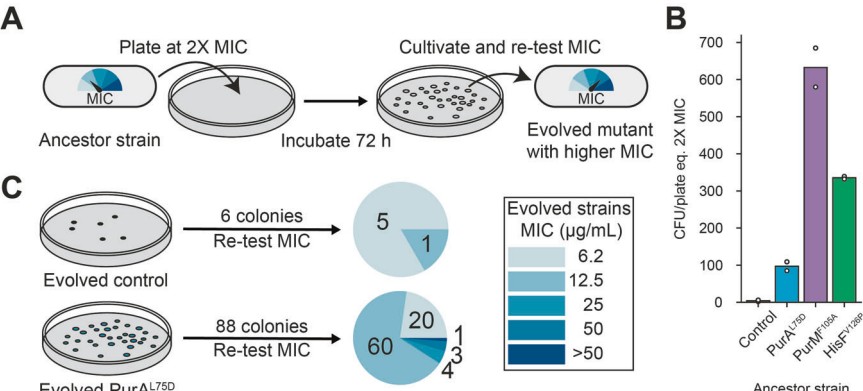

**Figure EV4.   Evolution of higher carbenicillin resistance.**

(**A**) Schematic of the experimental workflow to evolve higher carbenicillin resistance. (**B**) Number of spontaneous mutants that appeared after 3 days incubation of the control strains and the three mutants (HisF[V126P], PurM[F105A] and PurA[L75D]) at their respective 2× MIC (3.1 µg/mL for the control strain and 6.2 µg/mL for the purine mutants). Bars are means of $n = 2$ distinct samples (dots). (**C**) Pie charts show carbenicillin MIC values of the evolved control strains and the evolved PurA[L75D] strains. 88 colonies were picked from a plate inoculated with the PurA[L75D] strain. All 6 colonies on the plate with the control strain were picked. Agar dilution assays on minimal glucose agar were performed to assess the MIC of these strains.

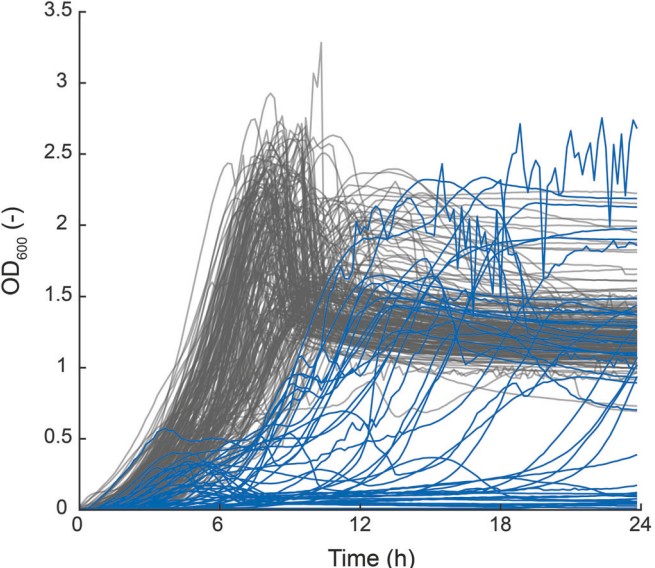

**Figure EV5. Growth of 235 clinical *E. coli* isolates on minimal glucose medium.**

Curves are the mean of $n = 2$ cultures in 96-well plates. Blue lines show 41 strains with the lowest area under the curve (AUC). These strains were used for metabolome analysis.

