## [Peer Review File · Molecular Systems Biology]

Metabolic mutations reduce antibiotic susceptibility of *E. coli* by pathway-specific bottlenecks

Paul Lubrano, Fabian Smollich, Thorben Schramm, Elisabeth Lorenz, Alejandra Alvarado, Seraina Eigenmann, Amelie Stadelmann, Sevvalli Thavapalan, Nils Waffenschmidt, Timo Glatter, Nadine Hoffmann, Jennifer Müller, Silke Peter, Knut Drescher, and Hannes Link

Corresponding author(s): Hannes Link (hannes.link@uni-tuebingen.de)

Review Timeline:

Submission Date:	2nd Jul 24
Editorial Decision:	1st Aug 24
Revision Received:	28th Oct 24
Editorial Decision:	22nd Nov 24
Revision Received:	9th Dec 24
Accepted:	12th Dec 24

Editor: Jingyi Hou

Transaction Report:

1st Aug 2024

Manuscript Number: MSB-2024-12503

Title: Metabolic mutations induce antibiotic resistance by pathway-specific bottlenecks

Author: Paul Lubrano

Fabian Schmollich

Thorben Schramm

Elisabeth Lorenz

Alejandra Alvarado

Seraina Eigenmann

Amelie Stadelmann

Sevalli Thavapalan

Nils Waffenschmidt

Timo Glatter

Nadine Hoffmann

Silke Peter

Knut Drescher

Hannes Link

Dear Prof Link,

Thank you for submitting your work to Molecular Systems Biology. We have now heard back from the three reviewers who agreed to evaluate your manuscript. As you will see from the reports below, the reviewers are overall supportive. They raise, however, a series of concerns, which we would ask you to address in a major revision.

The reviewers' recommendations are relatively clear, so there is no need to reiterate the points listed below. All the issues raised by the reviewers need to be satisfactorily addressed. As you may already know, our editorial policy allows in principle a single round of major revision, and it is therefore essential to provide responses to the reviewers' comments that are as complete as possible.

On a more editorial level, we would ask you to address the following issues:

- Please provide a .docx formatted version of the manuscript text (including legends for main figures, EV figures and tables). Please make sure that the changes are highlighted to be clearly visible.
- Please provide individual production quality figure files as .eps, .tif, .jpg (one file per figure).
- Please provide a .docx formatted letter INCLUDING the reviewers' reports and your detailed point-by-point responses to their comments. As part of the EMBO Press transparent editorial process, the point-by-point response is part of the Review Process File (RPF), which will be published alongside your paper.
- Please note that all corresponding authors are required to supply an ORCID ID for their name upon submission of a revised manuscript.
- We replaced Supplementary Information with Expanded View (EV) Figures and Tables that are collapsible/expandable online (see examples in <http://msb.embopress.org/content/11/6/812>). A maximum of 5 EV Figures can be typeset. EV Figures should be cited as 'Figure EV1, Figure EV2' etc... in the text and their respective legends should be included in the main text after the legends of regular figures.

Additional Tables/Datasets should be labeled and referred to as Table EV1, Dataset EV1, etc. Legends have to be provided in a separate tab in case of .xls files. Alternatively, the legend can be supplied as a separate text file (README) and zipped together with the Table/Dataset file.

For the figures and tables that you do NOT wish to display as Expanded View figures, they should be bundled together with their legends in a single PDF file called *Appendix*, which should start with a short Table of Content. Each legend should be below the corresponding Figure/Table in the Appendix. Appendix figures and tables should be referred to in the main text as: "Appendix Figure S1, Appendix Figure S2, Appendix Table S1" etc. See detailed instructions regarding expanded view here: <https://www.embopress.org/page/journal/17444292/authorguide#expandedview>.

- Before submitting your revision, primary datasets (and computer code, where appropriate) produced in this study need to be deposited in an appropriate public database (see <http://msb.embopress.org/authorguide> - dataavailability

<https://www.embopress.org/page/journal/17444292/authorguide#dataavailability>).

The accession numbers and database should be listed in a formal "Data Availability" section (placed after Materials & Method) that follows the model below (see also <https://www.embopress.org/page/journal/17444292/authorguide#dataavailability>). Please note that the Data Availability Section is restricted to new primary data that are part of this study.

Data availability

- RNA-Seq data: Gene Expression Omnibus GSE46843 (<https://www.ncbi.nlm.nih.gov/geo/query/acc.cgi?acc=GSE46843>)

- [data type]: [name of the resource] [accession number/identifier/doi] ([URL or identifiers.org/DATABASE:ACCESSION])

-At EMBO Press we ask authors to provide source data for the main figures. Our source data coordinator will contact you to discuss which figure panels we would need source data for and will also provide you with helpful tips on how to upload and organize the files.

- Our journal encourages inclusion of *data citations in the reference list* to directly cite datasets that were re-used and obtained from public databases. Data citations in the article text are distinct from normal bibliographical citations and should directly link to the database records from which the data can be accessed. In the main text, data citations are formatted as follows: "Data ref: Smith et al, 2001". In the Reference list, data citations must be labeled with "[DATASET]". A data reference must provide the database name, accession number/identifiers and a resolvable link to the landing page from which the data can be accessed at the end of the reference. Further instructions are available at .

- We updated our journal's competing interests policy in January 2022 and request authors to consider both actual and perceived competing interests. Please review the policy <https://www.embopress.org/competing-interests> and update your competing interests if necessary.

Please use the heading "Disclosure statement and competing interests".

- All Materials and Methods need to be described in the main text using our 'Structured Methods' format, which is required for all research articles. According to this format, the Methods section includes a Reagents and Tools Table (listing key reagents, experimental models, software and relevant equipment and including their sources and relevant identifiers) followed by a Methods and Protocols section describing the methods using a step-by-step protocol format. The aim is to facilitate adoption of the methodologies across labs. More information on how to adhere to this format as well as a downloadable template (.docx) for the Reagents and Tools Table can be found in our author guidelines:

<https://www.embopress.org/page/journal/17444292/authorguide#structuredmethods>.

- Regarding data quantification:

Please ensure to specify the name of the statistical test used to generate error bars and P values, the number (n) of independent experiments (please specify technical or biological replicates) underlying each data point and the test used to calculate p-values in each figure legend. Discussion of statistical methodology can be reported in the materials and methods section, but figure legends should contain a basic description of n, P and the test applied.

Graphs must include a description of the bars and the error bars (s.d., s.e.m.).

- Please provide a "standfirst text" summarizing the study in one or two sentences (approximately 250 characters, including space), three to four "bullet points" highlighting the main findings and a "synopsis image" (550px width and 400-600 px height, PNG format) to highlight the paper on our homepage.

Here are a couple of examples:

<https://www.embopress.org/doi/10.15252/msb.20199356>

<https://www.embopress.org/doi/10.15252/msb.20209475>

<https://www.embopress.org/doi/10.15252/msb.209495>

When you resubmit your manuscript, please download our CHECKLIST (<https://www.embopress.org/pb-assets/embosite/EMBO%20Press%20Author%20Checklist-1642513524327.xlsx>) and include the completed form in your submission.

Please note that the Author Checklist will be published alongside the paper as part of the transparent process (<https://www.embopress.org/page/journal/17444292/authorguide#transparentprocess>).

If you feel you can satisfactorily deal with these points and those listed by the referees, you may wish to submit a revised version of your manuscript. Please attach a covering letter giving details of the way in which you have handled each of the points raised by the referees. A revised manuscript will be once again subject to review and you probably understand that we can give you no

guarantee at this stage that the eventual outcome will be favorable.

I look forward to receiving your revised manuscript soon.

Kind regards,
Jingyi

Jingyi Hou, PhD
Scientific Editor
Molecular Systems Biology

We realize that it is difficult to revise to a specific deadline. In the interest of protecting the conceptual advance provided by the work, we recommend a revision within 3 months (30th Oct 2024). Please discuss the revision progress ahead of this time with the editor if you require more time to complete the revisions. Use the link below to submit your revision:

IMPORTANT: When you send your revision, we will require the following items:

1. the manuscript text in LaTeX, RTF or MS Word format
2. a letter with a detailed description of the changes made in response to the referees. Please specify clearly the exact places in the text (pages and paragraphs) where each change has been made in response to each specific comment given
3. three to four 'bullet points' highlighting the main findings of your study
4. a short 'blurb' text summarizing in two sentences the study (max. 250 characters)
5. a 'thumbnail image' (550px width and max 400px height, Illustrator, PowerPoint or jpeg format), which can be used as 'visual title' for the synopsis section of your paper.
6. Please include an author contributions statement after the Acknowledgements section (see <https://www.embopress.org/page/journal/17444292/authorguide>)
7. Please complete the CHECKLIST available at (<https://bit.ly/EMBOPressAuthorChecklist>). Please note that the Author Checklist will be published alongside the paper as part of the transparent process (<https://www.embopress.org/page/journal/17444292/authorguide#transparentprocess>).
8. When assembling figures, please refer to our figure preparation guideline in order to ensure proper formatting and readability in print as well as on screen:
<https://bit.ly/EMBOPressFigurePreparationGuideline>
See also figure legend guidelines: <https://www.embopress.org/page/journal/17444292/authorguide#figureformat>
9. Please note that corresponding authors are required to supply an ORCID ID for their name upon submission of a revised manuscript (EMBO Press signed a joint statement to encourage ORCID adoption). (<https://www.embopress.org/page/journal/17444292/authorguide#editorialprocess>)
Currently, our records indicate that the ORCID for your account is 0000-0002-6677-555X.

Link Not Available

11. Include a Reagents and Tools Table as part of the Methods section, which can be downloaded from our author guidelines (<https://www.embopress.org/page/journal/17444292/authorguide#structuredmethods>)

*** PLEASE NOTE *** As part of the EMBO Press transparent editorial process initiative (see our Editorial at <https://dx.doi.org/10.1038/msb.2010.72>), Molecular Systems Biology publishes online a Review Process File with each accepted manuscripts. This file will be published in conjunction with your paper and will include the anonymous referee reports, your point-by-point response and all pertinent correspondence relating to the manuscript. If you do NOT want this File to be published, please inform the editorial office at msb@embo.org within 14 days upon receipt of the present letter.

Reviewer #1:

The study "Metabolic mutations induce antibiotic resistance by pathway specific bottlenecks" by Lubrano, et al., addresses a particularly relevant and interesting topic: how introducing metabolic bottlenecks can affect antibiotic efficacy. The authors used an original approach in which they combine CRISPR introduced point mutations destabilizing essential proteins and mass spectrometry to monitor the effect of point mutations conferring resistance to gentamicin and carbenicillin.

I find the topic exciting and of broad interest, and the results can help the community working on antibiotic resistance and tolerance to better understand previous findings relating mutations in metabolic genes and evolution of resistance.

Here my main comments.

The authors convincingly showed that although mutations causing bottlenecks in metabolism don't contribute to a large change in resistance, their role in resistance goes beyond mere reduction of growth rate. The authors proposed that the major mechanisms by which mutations in genes related to respiratory chain and IMP/purine metabolism confer resistance to gentamicin and carbenicillin, respectively, is mainly through an indirect but specific change of drug uptake.

While intuitive for gentamicin, it is not yet clear to me how a bottleneck in purine metabolism would affect the uptake of carbenicillin, or why interfering with ATP metabolism would result in antibiotic-specific changes. Measuring uptake of the antibiotics in wildtype and mutants could strengthen these findings and provide solid evidence supporting authors' conclusions. Optionally, transcriptome analysis of mutants could further support indirect changes in the expression of genes encoding for transporters and gain potential insights on mechanisms of resistance.

While I find the profiling of clinical isolates an interesting resource, the conclusions drawn are less convincing. Several bottlenecks of clinical isolates found in vitro could simply reflect that, in vivo *E. coli* has access to many more nutrients than what is normally present in minimal glucose medium - i.e. mutations in PurK could be neutral mutations. Hence, whether mutations in PurK are the result of carbenicillin selective pressure is hard to conclude from the provided evidence. I would revise the text to avoid any overstatements.

Minor

Because tolerance is largely dependent on where the bacteria come from (e.g. stationary phase, starvation ...), it would help to have more details in the main text on how tolerance was evaluated.

I find sometimes the text contradictory. For example: Sentence line 75:78 vs sentence line 78:80

Line 130: it would be important to report p-value significance of how much enriched are mutations in metabolic genes based on the number of metabolic genes initially tested in the library.

Line 175: Here I would argue that growth defect would confer tolerance rather than resistance. This is also what was stated before by the authors. I would clarify this point.

Reviewer #2:

1. General Comments:

In this study, the authors employ a high-throughput CRISPR library screening to identify mutations that confer resistance to carbenicillin and gentamicin, revealing pathway-specific effects.

The key takeaway is that metabolic mutations in *Escherichia coli* mostly confer antibiotic resistance through pathway-specific mechanisms, instead of a general reduction in growth rate. Specifically, resistance to carbenicillin is predominantly associated with mutations in the purine nucleotide biosynthesis pathway, while resistance to gentamicin is linked to mutations in the respiratory chain. These findings suggest that metabolic bottlenecks induced by these mutations lead to low-level antibiotic resistance, which can promote the evolution of higher resistance levels over time. Similar metabolic bottlenecks were identified in clinical *E. coli* isolates, underscoring the clinical relevance of these metabolic pathways in contributing to antibiotic resistance. The study is well carried out and it uses a combination of different tools, including high-throughput screening enabled by NGS and subsequent validation. When made publicly available, the data arising from this work can also be broadly useful for the research community.

However, there are areas where the manuscript can be improved to enhance clarity and rigor.

Specific points:

1. The introduction could be improved by more clearly stating the aims and objectives earlier in the section.

2. I appreciate the use of the large-scale CRISPR mutant library. It's helpful to provide more details on the criteria used to select the target genes and target amino acids that were changed. I couldn't find this information in the paper.
3. Provide a more detailed explanation of the experimental design, particularly the rationale for choosing specific antibiotic concentrations and the criteria for defining resistance. For instance, the choice of 2X MIC for screening should be justified more clearly.
4. In this work, the authors used "resistance" and "tolerance" to characterize different mutants. Their usage is specifically tied to their experimental design. For example, putative resistance is characterized initially as a mutant with sufficient enrichment in sequencing reads (after being exposed to 2xMIC antibiotic). "Tolerance" is defined as slower killing (relative to the wild-type) when exposed to an antibiotic at a lethal concentration. Given the typical confusion surrounding the use of these terms, it's useful to more explicitly define them in the paper.
5. They used a dilution assay to test some putative mutants. From Figures EV2 and EV3, it appears that some mutants exhibit collective tolerance/resistance (or inoculum effect), whereby cells at higher inoculums survived better during antibiotic treatment. The authors might want to discuss these a bit more.
6. In Figure EV1, the data points appear to be quite sparse for gentamicin treatment. Also, the PCC between two replicate plates appears to be much lower than those from other treatment conditions. Can you explain this apparent discrepancy?

Reviewer #3:

Lubrano et al present a study about the link between metabolism and the susceptibility of *E. coli* against the antibiotics carbenicillin (carb) and gentamicin (genta). They use a system-wide screening approach with a previously established CRISPR library to identify partial loss-of-function mutations in metabolic genes that lead to decreases in antibiotic susceptibility. The authors perform a number of follow-up experiments including metabolomics, growth and kill experiments, and laboratory evolution with targeted CRISPR mutants, as well as genomics, metabolomics, and growth and kill experiments with clinical isolates. The main mechanisms identified were related to purine nucleotide biosynthesis for carb and the respiratory chain for genta susceptibility. The authors show that the mutations lead to metabolic bottlenecks and fitness defects. However, the ability to grow at increased antibiotic concentrations is apparently not related to general fitness defects, indicating that they are pathway specific. In contrast, the fitness defects generally mediate increased survival (tolerance). The manuscript tackles a topic that had gained recent interest in the AMR field. Overall, the manuscript contains a wealth of data and insights. The experiments are well executed, and the text is well written. The manuscript will be an important contribution to the AMR field.

Major points

The term resistance should be used carefully. While some authors defined resistance as a decrease in susceptibility (as measured by growth at increased antibiotic concentrations), most practitioners define resistance as decreased susceptibility in terms of growth above the clinical breakpoint in standard microbroth dilution assays (with MHII broth). The observed changes in MIC are rather small and the used method as well as the medium (for obvious reasons) do not align with the international standard. Therefore, the authors should use the term 'susceptibility' instead of resistance. I acknowledge that this would make the bulk of text a bit difficult to read. One way out would be to use the term 'susceptibility' in the title and the abstract. The authors can provide their definition of resistance in the introduction, and then use the term 'resistance' onwards.

Specific remarks

Abstract: The magnitude of the changes in MIC should be mentioned here in order to allow readers to directly judge the clinical relevance of the observed effects.

L34 link between tolerance and evolvability: Can it be really linked to tolerance only? In L274 the authors discuss that increases in growth and survival may promote evolution of high-level resistance. I agree to this. The effect of both, increases in growth and survival, on evolution cannot be disentangled. In addition, other mechanisms may play a role. Please amend the statement in the abstract.

L91 The authors should explain the nature of the screen better to for those, who are not 100% familiar with CRISPR screens. I found myself stumbling across the term 'repair template'. My understanding is that the plasmid with the repair template-sgRNA still resides in the cells without any function. Its function was (previously) to introduce the genomic mutation (the repair template contains the mutation). The presence of the genomic mutation in the host cell is inferred from the presence of the plasmid with the mutated repair template-sgRNA (barcode). However, reading the description of the screen and having the background information only in a referenced paper, I found myself wondering whether the function of the repair-template in the screen is actually repairing the introduced genomic mutation, getting active throughout the screen, and only those clones that have the repaired mutation are being detected. I think a few more words and a slight amendment of Fig. 1A would help to avoid any confusion of readers.

CRISPR screen: the authors state in L393 that they added kanamycin and chloramphenicol if the strains harbored pTS040 and pTS041. It is very difficult for readers to understand that those are the plasmids harbored by the strains in the pooled library. This should be clarified. To avoid ambiguity, the authors should clearly state any additional antibiotics contained in each medium for all the different assays described in the methods section. E.g. L515 Did the M9 agar plates contain Kan and Cm? If so, the authors should discuss any bias that might be introduced by the combination effect of Kan or Cm with Carb or Genta. E.g. Yeh et. al showed that beta-lactams (ampicillin) and aminoglycosides (tobramycin) have synergistic effects, while chloramphenicol and aminoglycosides (amikacin) can have antagonistic effects in *E. coli*.

L102 Did the authors sequence the mutants that grew on plates with higher conc.? The lower number of cells on plates with 6-fold MIC as compared to 2-fold MIC might indicate increased fitness costs linked to a stronger and more relevant decrease in susceptibility. If yes, I recommend to add this data to the manuscript.

L365 Discussion: The authors used defined M9 medium to unravel the effect of metabolic perturbations on resistance. The susceptibility was restored by adding e.g. purines. I am missing a discussion whether these changes will be detectable under standard conditions (MHII broth containing purines) and how those metabolic dependencies play out in the host environment. Are there in vivo situations, in which some of those metabolites are limiting, exposing the metabolism-dependent resistance mechanisms in vivo?

L187 '...because it influences drug transport (Prajapati et al, 2021) or penicillin binding protein specificity (Spratt, 1977)': The authors should define what they mean with the word 'it' - the beta-lactam structure or the resistance mechanism. If the resistance mechanisms, how can the authors infer this only from the MICs to different substances?

L240 and throughout: The discussion around specific and global effects is unclear. Also the specific effects, through metabolic bottlenecks, will lead to global effects. Therefore, it cannot be excluded that global effects play a role. Rather, the data allows the authors to state that global effects alone are not sufficient to explain the results. Still, global effects cannot be excluded to be required for the observed effects.

L272 Evolution experiment: Clarify the word 'prolonged'? I couldn't find a methodological description of the evolution experiment. Please add. It seems that it was an evolutionary rescue experiment. The authors should discuss the possible explanations for the increase in mutants; i) higher mutation rates in metabolic mutants (stress-induced mutagenesis), ii) different/increased evolutionary paths to resistance, or iii) population dynamics on the plate (increased survival and chance for phenotypic adaption, see <https://pubmed.ncbi.nlm.nih.gov/30647458/>).

Fig3C and L264: Clearly, the addition of adenine leads to a convergence of kill curves between wt and mutants. However, it seems that the tolerance of the *purA* and *hisF* mutants remains unchanged and the tolerance of the wt increases, as opposed to a reversal of the tolerance phenotype of all mutants. Please amend the statement.

L262 global effect on tolerance: Did the authors measure tolerance of a sufficient number of strains to make a correlation analysis between growth rate and tolerance? If yes, please provide.

Fig. 4: Descriptions of panels E,F, G are mixed up.

Method section: The authors should logically order of sub-sections within the methods section. CRISPR screen after media and link with Illumina sequencing; followed by preparation of CRISPR and CRISPRi mutants. Next, generation of growth curves followed by agar dilution assays and by time-kill curves. Next, metabolomic cultivations followed by metabolome measurements. Next, WGS of clinical strains. Statistics in the end.

L405 The authors should provide a statement how they interpreted the outcome in terms of MIC fold change. E.g. looking at Fig. 1B, one may either infer a MIC change of 1.5-fold or 2-fold. Which inoculum OD is the one that they used and what coverage is scored as growth (is a single colony enough or do they require a full spot?)

Not all co-authors are listed in the author contribution statement.

We thank the reviewers for their thoughtful and constructive comments. We addressed all points as described below. Our answers are blue and changes in the manuscript are *italic*. Major revisions and additional experiments include:

- We better distinguish between resistance and tolerance and use standard definitions from the literature. We now refer to the observed resistance phenotypes as “low-level resistance” or “reduced antibiotic susceptibility”, to address the valid concern about the use of the term “resistance” for 2X MIC increases.
- We performed RNA-seq with the PurA mutant and identified upregulation of transporters and stress-response genes. We added the transcriptome data in a **new Appendix Figure S4** and discuss it in the text.
- We performed additional experiments to explain the rationale for selecting specific antibiotic concentrations (2X MIC). Additional experiments at 2X, 4X, 6X, and 8X MIC were done for both carbenicillin and gentamicin to show the difference between CRISPR library and control strain (**New Appendix Figure S1**).
- We sequenced CRISPR strains from plates with higher MICs (6X for carbenicillin and 10X for gentamicin) and found that some mutants can indeed confer higher resistance levels (but only for gentamicin).

Response to Reviewer #1:

The study "Metabolic mutations induce antibiotic resistance by pathway specific bottlenecks" by Lubrano, et al., addresses a particularly relevant and interesting topic: how introducing metabolic bottlenecks can affect antibiotic efficacy. The authors used an original approach in which they combine CRISPR introduced point mutations destabilizing essential proteins and mass spectrometry to monitor the effect of point mutations conferring resistance to gentamicin and carbenicillin.

I find the topic exciting and of broad interest, and the results can help the community working on antibiotic resistance and tolerance to better understand previous findings relating mutations in metabolic genes and evolution of resistance.

We thank reviewer #1 for the positive and constructive comments.

Here my main comments.

The authors convincingly showed that although mutations causing bottlenecks in metabolism don't contribute to a large change in resistance, their role in resistance goes beyond mere reduction of growth rate. The authors proposed that the major mechanisms by which mutations in genes related to respiratory chain and IMP/purine metabolism confer resistance to gentamicin and carbenicillin, respectively, is mainly through an indirect but specific change of drug uptake.

While intuitive for gentamicin, it is not yet clear to me how a bottleneck in purine metabolism would affect the uptake of carbenicillin, or why interfering with ATP metabolism would result in antibiotic-specific changes. Measuring uptake of the antibiotics in wildtype and mutants could strengthen these findings and provide solid evidence supporting authors conclusions.

Optionally, transcriptome analysis of mutants could further support indirect changes in the expression of genes encoding for transporters and gain potential insights on mechanisms of resistance.

We agree with this point, and also wondered how purine mutations lead to MIC increases. To address this, we followed the suggestion of the reviewer and performed RNA-seq, and we also tried

to measure antibiotic uptake. With the uptake assays we encountered technical difficulties. Briefly, carbenicillin-treated cells formed filaments, which caused problems when we used filter membranes for washing cells to measure intracellular carbenicillin levels. We think the main problem was that the filters blocked and we could not obtain reproducible results. We therefore tried to measure drug uptake indirectly by assessing extracellular carbenicillin levels and how much it decreases in the culture medium of different strains. However, we observed no significant difference between the strains and the medium control without cells.

Therefore, we followed the suggestion to measure the transcriptome of our strains with RNA-seq. We measured the transcriptome of the PurA mutant strain and the control strain as a reference. This revealed several interesting changes:

- 1) Downregulation of purine genes likely due to activation of the PurR repressor by high levels of hypoxanthine (the activator metabolite of PurR).
- 2) Downregulation of the xanthine transporter XanP.
- 3) Upregulation of the inner membrane transporter *yjiY* and acid stress response genes (e.g. *gadA* and *hdeA*).

All of these responses might be linked to reduced ATP levels and changes in proton gradients. We show the transcriptome data in a **new Appendix Figure S4** and **Table EV4** and discuss the implications in the revised manuscript. In the discussion we discuss these data in the light of the many hypotheses in the literature regarding the role of purine metabolism mutations in antibiotic resistance.

In the result section:

*“In the PurA^{L75D} mutant, we observed high levels of inosine and hypoxanthine (Figure 2E), probably due to the accumulation of the PurA substrate IMP. Since hypoxanthine is as an allosteric activator of the purine repressor (PurR), we expected a downregulation of purine biosynthesis genes. Consistent with this, the transcriptome of the PurA^{L75D} strain showed a significant downregulation of genes in purine nucleotide biosynthesis, which may aggravate the bottleneck in the pathway (Appendix Figure S4 and Table EV4). The strongest transcriptome response in the PurA^{L75D} strain was a downregulation of the xanthine transporter *xanP* and an upregulation of the pyruvate transporter *yjiY*. Moreover, genes associated with acid stress response were upregulated (e.g. *gadABC* and *hdeABD*). These transcriptional changes, individually or in combination, may contribute to the MIC increases in the PurA^{L75D} strain, potentially by altering transport processes or stress responses.”*

In the discussion:

*“Although *OmpF* expression did not show changes at the transcriptional level (Table EV4), its regulation is known to occur post-transcriptionally through small regulatory RNAs.”*

While I find the profiling of clinical isolates an interesting resource, the conclusions drawn are less convincing. Several bottlenecks of clinical isolates found in vitro could simply reflect that, in vivo *E. coli* has access to many more nutrients than what normally present in minimal glucose medium - i.e. mutations in PurK could be neutral mutations. Hence, whether mutations in PurK are the results of carbenicillin selective pressure is hard to conclude from the provided evidence. I would revise the text to avoid any overstatements.

We agree that these mutations are probably accidents due to conditions where the respective pathway is not required. Therefore, we removed any claim that there is a selective pressure for metabolic mutations and revised the discussion of these results:

“We cannot conclude that the acquisition of the PurK^{E49G} mutation is the result of β -lactam selective pressure. Instead, it is more likely that PurK^{E49G} was acquired in an environment where de novo purine synthesis is not essential. For example, human urine has been shown to complement gene deletions of genes in purine biosynthesis (Ma et al, 2018), and E. coli can salvage nucleotides present in urine (Andersen-Civil et al, 2018).”

Minor

Because tolerance is largely dependent on where the bacteria come from (e.g. stationary phase, starvation ...), it would help to have more details in the main text on how tolerance was evaluated.

We clarified that all time-kill assays were performed with exponentially growing cultures that had an OD₆₀₀ of at least 0.2 and not more than 0.8. We added this information in the main text and in the figure captions:

Main text: *“The purine mutants, the LeuB^{I134P} strain and the control strain were cultivated in minimal medium until they reached exponential phase and optical densities (OD) between 0.2 and 0.8.”*

Figure Captions: *“Before treatment with carbenicillin/sulbactam both cultures reached exponential phase and optical densities (OD) between 0.2 and 0.8.”*

I find sometimes the text contradictory. For example: Sentence line 75:78 vs sentence line 78:80

We agree that the distinction between resistance and tolerance was not always clear. In the revised text we tried to better separate the concepts of pathway-specific resistance and growth rate-dependent tolerance. For example, in the section mentioned by the reviewer:

“These results demonstrate that metabolic mutations confer antibiotic resistance in a pathway-specific manner, rather than by a general reduction in growth rate or an overall metabolic state. However, while the same metabolic mutations that conferred resistance to carbenicillin also led to antibiotic tolerance, this tolerance was mainly a result of the reduced growth rates of the metabolic mutants, which is consistent with previous studies (Lee et al, 2018).”

Line 130: it would be important to report pvalue significance of how much enriched are mutations in metabolic genes based on the number of metabolic genes initially tested in the library.

We have performed a Chi-square to test the enrichment of metabolic mutations. For both antibiotics metabolic mutations are significantly enriched over non-metabolic mutation (p-values below 0.001). We added this information in the text and a **new Table EV2**:

“Thus, our CRISPR screen identified potential resistance mutations, with a significant enrichment of mutations in metabolic genes compared to non-metabolic genes (Table EV2).”

Line 175: Here I would argue that growth defect would confer tolerance rather than resistance. This is also what stated before by the authors. I would clarify this point.

We revised this section to clarify this point:

“This implies that the putative resistance phenotypes are pathway-specific and that reduced growth alone is not sufficient to confer resistance.”

Response to Reviewer #2:

1. General Comments:

In this study, the authors employ a high-throughput CRISPR library screening to identify mutations that confer resistance to carbenicillin and gentamicin, revealing pathway-specific effects.

The key takeaway is that metabolic mutations in *Escherichia coli* mostly confer antibiotic resistance through pathway-specific mechanisms, instead of a general reduction in growth rate. Specifically, resistance to carbenicillin is predominantly associated with mutations in the purine nucleotide biosynthesis pathway, while resistance to gentamicin is linked to mutations in the respiratory chain. These findings suggest that metabolic bottlenecks induced by these mutations lead to low-level antibiotic resistance, which can promote the evolution of higher resistance levels over time. Similar metabolic bottlenecks were identified in clinical *E. coli* isolates, underscoring the clinical relevance of these metabolic pathways in contributing to antibiotic resistance.

The study is well carried out and it uses a combination of different tools, including high-throughput screening enabled by NGS and subsequent validation. When made publicly available, the data arising from this work can also be broadly useful for the research community.

However, there are areas where the manuscript can be improved to enhance clarity and rigor.

We thank reviewer #2 for the positive and constructive comments.

Specific points:

1. The introduction could be improved by more clearly stating the aims and objectives earlier in the section.

We agree with this point and stated the objective and aim, which was motivated by the general observation that slowing down essential processes can influence antibiotic action. We added this in the introduction:

*“The aim of this study was to test if slowing down essential processes can confer resistance to antibiotics and to systematically identify these processes. Therefore, we used a CRISPR library of *E. coli* strains, each carrying a point mutation in an essential gene, many of which are likely to reduce their activity. By measuring the antibiotic susceptibility of these strains, we sought to determine whether mutations in essential genes can increase the minimum inhibitory concentration (MIC) of antibiotics.”*

2. I appreciate the use of the large-scale CRISPR mutant library. It's helpful to provide more details on the criteria used to select the target genes and target amino acids that were changed. I couldn't find this information in the paper.

We agree that this information was missing and was only given in our previous study (Schramm et al 2023). Therefore, we summarize the results from Schramm et al. and included it in the revised manuscript:

*“To test this hypothesis, we measured antibiotic resistance of an *E. coli* CRISPR library that we constructed in a previous study (Schramm et al, 2023). In that study, we selected 352 proteins that are essential for *E. coli* growth on minimal glucose medium. For each protein, we designed up to 50 amino acid changes (10 sites, with 5 substitutions per site). For 154 proteins, fewer than 50 substitutions were feasible due to limitations by our design rules for CRISPR gene editing. The mutations were designed to destabilize the proteins and were primarily located at buried sites. After*

gene editing, the library contained 15,120 of the 16,038 designed single amino acid substitutions and targeted 346 genes. Most mutations (8,290) were classified as “low-fitness” mutations (Schramm et al., 2023) and these mutations are the focus of this study, because they may slow down the associated essential process.”

3. Provide a more detailed explanation of the experimental design, particularly the rationale for choosing specific antibiotic concentrations and the criteria for defining resistance. For instance, the choice of 2X MIC for screening should be justified more clearly.

We agree that this is an important point. Initially we selected this concentration based on a pilot experiment with carbenicillin, where we plated the CRISPR library and the control strain and counted CFUs at different carbenicillin concentrations. Stronger differences in CFUs between the control and the library were only visible at 2X MIC, as shown in Figure S1 of the previous manuscript version.

We have now performed additional experiments at 2X, 4X, 6X, and 8X MIC for both gentamicin and carbenicillin, with replicates, to further support the rationale for selecting 2X MIC for the screen. This confirmed that our mutants cannot achieve more than 2X MIC increases for carbenicillin. But for gentamicin there are mutants with up to 10X MIC increases. We added this in **a new Appendix Figure S1**.

“First, we grew the CRISPR library and the control strain on agar plates containing 2X, 4X, 6X, and 8X the MIC of carbenicillin (a β -lactam) and gentamicin (an aminoglycoside). For carbenicillin, we observed a marked difference in colony-forming units (CFUs) between the control strain and the library at 2X MIC, but this difference disappeared at 4X MIC and above (Appendix Figure S1). This suggests that most mutations in the library confer only low-level resistance to carbenicillin. For gentamicin, CFU differences between the CRISPR library and the control strain were still noticeable up to 8X MIC (Appendix Figure S1), but we used 2X MIC for both antibiotics in the subsequent screen for resistant mutants.”

4. In this work, the authors used "resistance" and "tolerance" to characterize different mutants. Their usage is specifically tied to their experimental design. For example, putative resistance is characterized initially as a mutant with sufficient enrichment in sequencing reads (after being exposed to 2xMIC antibiotic). "Tolerance" is defined as slower killing (relative to the wild-type) when exposed to an antibiotic at a lethal concentration. Given the typical confusion surrounding the use of these terms, it's useful to more explicitly define them in the paper.

We agree that there is confusion about these terms and in the revised manuscript we explicitly define antibiotic resistance and tolerance, according to: *Brauner A, Fridman O, Gefen O & Balaban NQ (2016) Distinguishing between resistance, tolerance and persistence to antibiotic treatment. Nat Rev Microbiol 14: 320–330.*

We added the definitions in the introduction:

“Antibiotic resistance and antibiotic tolerance are two distinct mechanisms by which bacteria evade the effects of antibiotics (Brauner et al, 2016). Resistance refers to the inherited ability of bacteria to grow at high antibiotic concentrations. Tolerance, in contrast, is the ability to survive exposure to high concentrations of an antibiotic without a corresponding increase in MIC. Tolerance is often linked to slowing down of essential bacterial processes (Brauner et al, 2016), but how essential processes influence antibiotic resistance remains unclear.”

And also, in the result section:

*“Apart from antibiotic resistance, antibiotic tolerance has been associated with slow growth, especially in the case of β -lactams (Lee et al, 2018). **Antibiotic tolerance has been defined as the ability to survive transient exposure to an antibiotic without change in the MIC (Brauner et al, 2016).**”*

We also define that antimicrobial resistance observed in our study, particularly at 2X MIC, does not meet the 'high concentration' criteria (Brauner et al, 2016). Therefore, we revised the text and make clear that the phenotypes we observed represent 'low-level' resistance, or as suggested by reviewer 3 reduced “antibiotic susceptibility”. We changed this for example in a new title and in the introduction:

“Although these mutants do not meet the strict definition of resistance, as they do not grow at high drug concentrations, we will refer to them as resistance mutations in the following.”

5. They used a dilution assay to test some putative mutants. From Figures EV2 and EV3, it appears that some mutants exhibit collective tolerance/resistance (or inoculum effect), whereby cells at higher inoculums survived better during antibiotic treatment. The authors might want to discuss these a bit more.

A paragraph was added to discuss this observation:

“Resistance was tested with agar dilution assays using several inoculum densities (Wiegand et al, 2008). In some cases, we found an inoculum effect where cells grew at high inoculum density (0.1 OD_{600}) and showed no growth or only single colonies at low inoculum density (0.0001 OD_{600}). This behavior could be caused by an artificial reduction of antibiotic concentration or because cells grow on a layer of dead cells. Therefore, we based our MIC determination only on the spots at the lower inoculum density (0.0001 OD_{600}).”

And also in the method section:

“MIC was determined based on the first concentration at which a full spot of cells does not grow at OD_{600} 0.0001 (Wiegand et al, 2008).”

6. In Figure EV1, the data points appear to be quite sparse for gentamicin treatment. Also, the PCC between two replicate plates appears to be much lower than those from other treatment conditions. Can you explain this apparent discrepancy?

The data points are indeed much sparser, because we detected fewer mutants with high read counts on gentamicin. The reason is that on the gentamicin plates more distinct colonies formed than on the carbenicillin plates (on which we observe many very small colonies). We think the reason is that gentamicin resistant mutants tend to have higher MICs and therefore form more distinct colonies. This is visible on pictures of the plates that we show in Figure EV1 (also shown below as Revision Figure 1). The larger colonies on gentamicin compared to carbenicillin are probably the reason for higher read counts in fewer mutants and the sparse data points and lower PCC in Figure EV1.

CRISPR library
0.45 µg/mL Gentamicin

CRISPR library
3.1 µg/mL Carbenicillin

Revision Figure 1 – Colonies from the CRISPR library at 2X gentamicin and 2X carbenicillin. On gentamicin there are more distinct colonies compared to carbenicillin, which may lead to higher read counts for fewer mutants and the sparse data points in Figure EV1.

Response to Reviewer #3:

Lubrano et al present a study about the link between metabolism and the susceptibility of *E. coli* against the antibiotics carbenicillin (carb) and gentamicin (genta). They use a system-wide screening approach with a previously established CRISPR library to identify partial loss-of-function mutations in metabolic genes that lead to decreases in antibiotic susceptibility. The authors perform a number of follow-up experiments including metabolomics, growth and kill experiments, and laboratory evolution with targeted CRISPR mutants, as well as genomics, metabolomics, and growth and kill experiments with clinical isolates. The main mechanisms identified were related to purine nucleotide biosynthesis for carb and the respiratory chain for genta susceptibility. The authors show that the mutations lead to metabolic bottlenecks and fitness defects. However, the ability to grow at increased antibiotic concentrations is apparently not related to general fitness defects, indicating that they are pathway specific. In contrast, the fitness defects generally mediate increased survival (tolerance). The manuscript tackles a topic that had gained recent interest in the AMR field. Overall, the manuscript contains a wealth of data and insights. The experiments are well executed, and the text is well written. The manuscript will be an important contribution to the AMR field.

We thank reviewer #3 for the positive and constructive comments.

Major points

The term resistance should be used carefully. While some authors defined resistance as a decrease in susceptibility (as measured by growth at increased antibiotic concentrations), most practitioners define resistance as decreased susceptibility in terms of growth above the clinical breakpoint in standard microbroth dilution assays (with MHII broth). The observed changes in MIC are rather small and the used method as well as the medium (for obvious reasons) do not align with the international

standard. Therefore, the authors should use the term 'susceptibility' instead of resistance. I acknowledge that this would make the bulk of text a bit difficult to read. One way out would be to use the term 'susceptibility' in the title and the abstract. The authors can provide their definition of resistance in the introduction, and then use the term 'resistance' onwards.

We agree with this point and followed the reviewer's suggestions.

1) We changed in the title "induce antibiotic resistance" for "reduce antibiotic susceptibility":

"Metabolic mutations reduce antibiotic susceptibility of E. coli by pathway-specific bottlenecks"

2) We use "reduced antibiotic susceptibility" throughout the revised abstract and introduction. At the end of the introduction we provide our definition of resistance:

"Although these mutants do not meet the strict definition of resistance, as they do not grow at high drug concentrations, we will refer to them as resistance mutations in the following."

Specific remarks

Abstract: The magnitude of the changes in MIC should be mentioned here in order to allow readers to directly judge the clinical relevance of the observed effects.

We added the following sentence to the abstract:

"Across all mutants, we observed only modest increases of the minimal inhibitory concentration (2- to 10-fold) without any cases of major resistance."

L34 link between tolerance and evolvability: Can it be really linked to tolerance only? In L274 the authors discuss that increases in growth and survival may promote evolution of high-level resistance. I agree to this. The effect of both, increases in growth and survival, on evolution cannot be disentangled. In addition, other mechanisms may play a role. Please amend the statement in the abstract.

We agree with this comment and removed the evolvability in the abstract to avoid confusion.

"Additionally, metabolic mutations conferred tolerance to carbenicillin by reducing growth rates."

In the main text, we added the statement that multiple factors can contribute to evolvability:

"However, multiple mechanism may contribute to this evolvability, including higher tolerance and resistance, stress responses (Pribis et al, 2022) or changes in mutation rates (Windels et al, 2019)."

L91 The authors should explain the nature of the screen better to for those, who are not 100% familiar with CRISPR screens. I found myself stumbling across the term 'repair template'. My understanding is that the plasmid with the repair template-sgRNA still resides in the cells without any function. Its function was (previously) to introduce the genomic mutation (the repair template contains the mutation). The presence of the genomic mutation in the host cell is inferred from the presence of the plasmid with the mutated repair template-sgRNA (barcode). However, reading the description of the screen and having the background information only in a referenced paper, I found myself wondering whether the function of the repair-template in the screen is actually repairing the introduced genomic mutation, getting active throughout the screen, and only those clones that have

the repaired mutation are being detected. I think a few more words and a slight amendment of Fig. 1A would help to avoid any confusion of readers.

This was indeed confusing, and we clarified this, because the repair template does not have a function throughout the screen, it is only active during the initial gene editing when the CRISPR system introduces the mutation into the genome (when we induce Cas9 expression and lambda red expression with aTc and arabinose). Once the genomic mutation is introduced and the editing process is completed, the plasmid with the repair template and sgRNA can remain in the cell, but it no longer plays a role. The plasmid is only used as a marker or “barcode” to identify the different mutants.

To address this point, we revised the text and Figure 1A:

“The repair template is used only once to introduce the mutation during the genome editing step. Afterwards, the genomic mutation remains stable in the E. coli cells, which we confirmed by sequencing of genomic DNA of selected mutants. Screening of all mutants in the library relies on sequencing of the pTS040 plasmid with the repair template and the sgRNA, which serve as a strain-specific barcode (Figure 1A).”

CRISPR screen: the authors state in L393 that they added kanamycin and chloramphenicol if the strains harbored pTS040 and pTS041. It is very difficult for readers to understand that those are the plasmids harbored by the strains in the pooled library. This should be clarified. To avoid ambiguity, the authors should clearly state any additional antibiotics contained in each medium for all the different assays described in the methods section. E.g. L515 Did the M9 agar plates contain Kan and Cm? If so, the authors should discuss any bias that might be introduced by the combination effect of Kan or Cm with Carb or Genta. E.g. Yeh et. al showed that beta-lactams (ampicillin) and aminoglycosides (tobramycin) have synergistic effects, while chloramphenicol and aminoglycosides (amikacin) can have antagonistic effects in E. coli.

We revised the manuscript and explain that kanamycin and chloramphenicol were added to all cultures as selection markers for pTS040 and pTS041. We explain this in the revised text and a new Figure 1A:

“Each strain in the library carries two plasmids (Figure 1A). The first plasmid (pTS040) has a repair template and a strain-specific sgRNA for gene editing, and a kanamycin resistance marker for selection. The second plasmid (pTS041) has the lambda Red genes and the Streptococcus pyogenes cas9 gene, with a chloramphenicol resistance marker.”

The M9 plated contained kanamycin and chloramphenicol, and we revised the method section to make this clearer:

“Kanamycin (50 µg/mL; Roth #T832.3) and chloramphenicol (30 µg/mL; Merck #C0378-25G) were added in either liquid or agar medium when strains harbored pTS040 (chloramphenicol resistance marker) and pTS041 (kanamycin resistance marker).”

We revised the text to explain that the control strain should account for biases from the selection markers:

“To avoid any bias from the antibiotics used for plasmid selection (kanamycin and chloramphenicol), we used a control strain containing the same plasmids (pTS040 and pTS041) and cultivated it under the same conditions as the CRISPR library, where kanamycin and chloramphenicol were added to all cultures.”

To further exclude biases, we grew the wild-type *E. coli* BW25113 without Cm and Kan and the control strain with Cm and Kan and observed similar sensitivity to carbenicillin and gentamicin. The antibiotic concentrations with no growth are the same for both strains as shown below in **Revision Figure 2**.

Revision Figure 2 – Growth curves of the control strain (BW25113 with pTS40 and pTS41) in M9 medium with kanamycin and chloramphenicol (left). Growth curves of the wild-type (WT) *E. coli* BW25113 in M9 glucose medium (right). In A carbenicillin was added at the concentration indicated in the legend. In B gentamicin was added. Lines are the mean of 6 replicates.

L102 Did the authors sequence the mutants that grew on plates with higher conc.? The lower number of cells on plates with 6-fold MIC as compared to 2-fold MIC might indicate increased fitness costs linked to a stronger and more relevant decrease in susceptibility. If yes, I recommend to add this data to the manuscript.

We performed additional experiments and sequenced mutants at higher MICs. Because only few colonies grow at these MICs we sequenced the plasmids individually (96 individual plasmids for both antibiotics). The results are included in the revised manuscript and in Table EV3:

“To further understand the range of MIC increases, we sequenced 96 individual colonies from plates with 6X MIC of carbenicillin, as well as 96 colonies from plates with 10X MIC of gentamicin (Table EV3). For carbenicillin, only 12 out of 96 strains matched the library barcodes, whereas 51 out of 96 strains matched for gentamicin. Most gentamicin-resistant mutations overlapped with those identified at 2X MIC, suggesting that metabolic mutations can confer higher resistance levels. In contrast, for carbenicillin, only one mutation, HisA^{V224Q}, overlapped with the 2X MIC mutants,

indicating that achieving higher resistance to carbenicillin via mutations in our library is difficult. These results are consistent with the CFU counts at higher MICs for the CRISPR library and the control strain (Appendix Figure S1)."

L365 Discussion: The authors used defined M9 medium to unravel the effect of metabolic perturbations on resistance. The susceptibility was restored by adding e.g. purines. I am missing a discussion whether these changes will be detectable under standard conditions (MHII broth containing purines) and how those metabolic dependencies play out in the host environment. Are there in vivo situations, in which some of those metabolites are limiting, exposing the metabolism-dependent resistance mechanisms in vivo?

We added information and references about purine metabolism in the host environment in the discussion:

"For example, human urine has been shown to complement gene deletions of genes in purine biosynthesis (Ma et al, 2018), and E. coli can salvage nucleotides present in urine (Andersen-Civil et al, 2018). Nevertheless, the de novo purine pathway has been shown to be essential for survival and colonization in niches such as the gut (Vogel-Scheel et al, 2010), human blood (Samant et al, 2008), or inside host cells (Shaffer et al, 2017)."

Additionally, we explain why we used M9 minimal medium and not MHII:

"We used minimal media to control the supply of metabolites and to systematically evaluate the response of E. coli to the availability of nutrients like adenine. This is not feasible with complex media like Mueller-Hinton broth, which contains variable components that may deplete unevenly during an experiment."

L187 '...because it influences drug transport (Prajapati et al, 2021) or penicillin binding protein specificity (Spratt, 1977)': The authors should define what they mean with the word 'it' - the beta-lactam structure or the resistance mechanism. If the resistance mechanisms, how can the authors infer this only from the MICs to different substances?

We made clear that this refers to the beta-lactam structure:

"The purine mutants were also resistant against aztreonam but not to meropenem (Appendix Figure S3), suggesting that the resistance mechanism is linked to the structure of the respective beta-lactam, because the β -lactam structure influences drug transport (Prajapati et al, 2021) or penicillin binding protein specificity (Spratt, 1977)."

L240 and throughout: The discussion around specific and global effects is unclear. Also the specific effects, through metabolic bottlenecks, will lead to global effects. Therefore, it cannot be excluded that global effects play a role. Rather, the data allows the authors to state that global effects alone are not sufficient to explain the results. Still, global effects cannot be excluded to be required for the observed effects.

We agree that the local affect alone might be insufficient and added this in the discussion:

"While our results show that resistance is due to bottlenecks in specific metabolic pathways, such as purine biosynthesis and respiratory chain-related processes, this does not exclude the possibility that global growth effects contribute to the resistance phenotype. However, our data suggest that global

effects alone, such as reduced growth rates or reduced fitness, are insufficient to induce resistance phenotypes.”

L272 Evolution experiment: Clarify the word 'prolonged'? I couldn't find a methodological description of the evolution experiment. Please add. It seems that it was an evolutionary rescue experiment. The authors should discuss the possible explanations for the increase in mutants; i) higher mutation rates in metabolic mutants (stress-induced mutagenesis), ii) different/increased evolutionary paths to resistance, or iii) population dynamics on the plate (increased survival and chance for phenotypic adaptation, see <https://pubmed.ncbi.nlm.nih.gov/30647458/>).

We added the information to the revised text (we incubated the plates 72 hours) and added phenotypic adaptation as a mechanism:

*“as evidenced by the >10X MIC increases found in the strains evolved from the PurA^{L75D} mutant **after 72h** exposure to carbenicillin (Figure EV4). However, multiple mechanism may contribute to this evolvability, including higher tolerance and resistance, stress responses (Pribis et al, 2022) or changes in mutation rates (Windels et al, 2019).”*

Moreover, we added a section that describes the evolution experiment in the methods section.

Fig3C and L264: Clearly, the addition of adenine leads to a convergence of kill curves between wt and mutants. However, it seems that the tolerance of the purA and hisF mutants remains unchanged and the tolerance of the wt increases, as opposed to a reversal of the tolerance phenotype of all mutants. Please amend the statement.

We agree with this point and we amended the statement:

“Nevertheless, the tolerance phenotype was metabolism-dependent, because with the addition of adenine the kill curves of the control strain and the purine mutants converged (Figure 3C). While adenine increased killing in the mutants, it decreased killing in the control strain, suggesting that adenine reduces tolerance of the mutants, but enhances survival of the control strain.”

L262 global effect on tolerance: Did the authors measure tolerance of a sufficient number of strains to make a correlation analysis between growth rate and tolerance? If yes, please provide.

The strains shown in the manuscript are the only ones we measured. We agree that we do not provide enough data to make a general statement for a “correlation”. We toned down the text but are confident that, with the support of published literature (e.g. Lee et al. 2018), the tolerance of our strains is at least in part due to their slow growth.

Fig. 4: Descriptions of panels E,F, G are mixed up.

Corrected

Method section: The authors should logically order of sub-sections within the methods section. CRISPR screen after media and link with Illumina sequencing; followed by preparation of CRISPR and CRISPRi mutants. Next, generation of growth curves followed by agar dilution assays and by time-kill curves. Next, metabolomic cultivations followed by metabolome measurements. Next, WGS of clinical strains. Statistics in the end.

We followed the reviewers suggestion and updated the ordering.

L405 The authors should provide a statement how they interpreted the outcome in terms of MIC fold change. E.g. looking at Fig. 1B, one may either infer a MIC change of 1.5-fold or 2-fold. Which inoculum OD is the one that they used and what coverage is scored as growth (is a single colony enough or do they require a full spot?)

This information was indeed missing. We always used the lower inoculum density (0.0001 OD₆₀₀) and we explain this in the revised text:

Results section:

“Resistance was tested with agar dilution assays using several inoculum densities (Wiegand et al, 2008). In some cases, we found an inoculum effect where cells grew at high inoculum density (0.1 OD₆₀₀), and showed no growth or only single colonies at low inoculum density (0.0001 OD₆₀₀). This behavior could be caused by an artificial reduction of antibiotic concentration or because cells grow on a layer of dead cells. Therefore, we based our MIC determination only on the spots at the lower inoculum density (0.0001 OD₆₀₀).”

Method section:

“MIC was determined based on the first concentration at which a full spot of cells does not grow at OD₆₀₀ 0.0001 (Wiegand et al, 2008).”

Not all co-authors are listed in the author contribution statement.

Corrected. All authors contributed to some of parts of the investigations and methods, and we list this as *all authors*:

*“Authors contribution: **Investigation: all authors; Methodology: all authors**; Conceptualization: PL, HL; Visualization: PL, FS, HL; Funding acquisition: PL, HL; Project administration: HL; Supervision: HL; Writing – original draft: PL, HL.”*

22nd Nov 2024

Manuscript Number: MSB-2024-12503R

Title: Metabolic mutations reduce antibiotic susceptibility of *E. coli* by pathway-specific bottlenecks

Author: Paul Lubrano

Fabian Smollich

Thorben Schramm

Elisabeth Lorenz

Alejandra Alvarado

Seraina Eigenmann

Amelie Stadelmann

Sevalli Thavapalan

Nils Waffenschmidt

Timo Glatter

Nadine Hoffmann

Jennifer Müller

Silke Peter

Knut Drescher

Hannes Link

Dear Prof Link,

Thank you for the submission of your revised manuscript to Molecular Systems Biology. We have now received the enclosed reports from the three reviewers who agreed to re-assess it. As you will see, the reviewers are overall supportive, and I am pleased to inform you that we will be able to accept your manuscript pending the following amendments:

1. Please address the remaining concerns from Reviewer #3.

On a more editorial level:

1. Please remove the figures from the manuscript file.

2. Please remove the "Author contribution" section from the manuscript file.

3. Please complete the author checklist.

4. Appendix: please change the title from "Appendix Figures" to "Appendix for" before the manuscript title; page numbers should be included in Table of Content.

5. Expanded view tables and datasets:

- Table EV1, EV3-EV6 should be renamed to Dataset EV# in source file names, titles, legends and manuscript callouts all need to be updated to Dataset EV1-EV#;

- Table EV2 and EV7 should remain EV tables, just renumbered to Table EV1-EV2 in source file names. Titles, legends and manuscript callouts should be updated to Table EV1-EV2.

- Legends should be included in a separate tab/sheet in each Excel file.

6. Please provide a "standfirst text" summarizing the study in one or two sentences (approximately 250 characters, including space), three to four "bullet points" highlighting the main findings and a "synopsis image" (550px width and 400-600 px height, PNG format) to highlight the paper on our homepage.

Here are a couple of examples:

<https://www.embopress.org/doi/10.15252/msb.20199356>

<https://www.embopress.org/doi/10.15252/msb.20209475>

<https://www.embopress.org/doi/10.15252/msb.209495>

7. Please indicate the statistical test used for data analysis in the legends of figure 2e.

8. During our standard image check, we noticed a potential figure reuse between Figure 2D and Appendix Figure S3C (specifically in the left most columns, see attached figure check report). It is somewhat strange as it appears that only the far-right cells within these columns seem to be reused (see attached screenshot). Could you please clarify this? Additionally, please provide the source data for the affected panels in both figures.

9. Source data files need to be saved in a scheme one figure/folder and then uploaded as .zip files. E.g. all the Source data files for figure 1 need to be saved in a single folder and this needs to be zipped and then uploaded as "SD figure 1.zip" file.

When you resubmit your manuscript, please download our CHECKLIST (<https://bit.ly/EMBOPressAuthorChecklist>) and include the completed form in your submission. *Please note* that the Author Checklist will be published alongside the paper as part of the transparent process (<https://www.embopress.org/page/journal/17444292/authorguide#transparentprocess>)

Click on the link below to submit your revised paper.

Thank you for submitting this exciting and interesting paper to Molecular Systems Biology.

Kind regards,
Jingyi

Jingyi Hou, PhD
Scientific Editor
Molecular Systems Biology

If you do choose to resubmit, please click on the link below to submit the revision online before 22nd Dec 2024.

Reviewer #1:

The authors have addressed all my comments and I support the publication of the manuscript

Reviewer #2:

The authors have fully addressed the issues I raised. However, it's not clear whether and how the data will be made publicly available.

Reviewer #3:

The authors carefully addressed most of the comments. However, there are two follow-ups detailed below.

L91 'Although these mutants do not meet the strict definition of resistance, as they do not grow at high drug concentrations, we will refer to them as resistance mutations in the following.'

Better: Although these mutants do not meet the definition of clinical resistance, because they do not grow above the clinical breakpoint concentrations when tested under standard conditions, we will refer to them as resistance mutations for simplicity in the following.

Initial reviewer comment:

CRISPR screen: the authors state in L393 that they added kanamycin and chloramphenicol if the strains harbored pTS040 and pTS041. It is very difficult for readers to understand that those are the plasmids harbored by the strains in the pooled library. This should be clarified. To avoid ambiguity, the authors should clearly state any additional antibiotics contained in each medium for all the different assays described in the methods section. E.g. L515 Did the M9 agar plates contain Kan and Cm? If so, the authors should discuss any bias that might be introduced by the combination effect of Kan or Cm with Carb or Genta. E.g. Yeh et. al showed that beta-lactams (ampicillin) and aminoglycosides (tobramycin) have synergistic effects, while chloramphenicol and aminoglycosides (amikacin) can have antagonistic effects in E. coli.

Reply:

L126 'To avoid any bias from the antibiotics used for plasmid selection (kanamycin and chloramphenicol), we used a control strain containing the same plasmids (pTS040 and pTS041) and cultivated it under the same conditions as the CRISPR library, where kanamycin and chloramphenicol were added to all cultures.'

The provided reply by the authors did not sufficiently exclude nor discuss the possibility of combination effects. This is important because the authors measure relatively small changes in MIC, which could be caused by such combination effects. The provided data is not included in the manuscript and does not determine the combination effect in a clean manner. One possibility would be to determine fractional inhibitory concentration index (FICI) (e.g. <https://academic.oup.com/jac/article/79/9/2394/7713050> as a start). However, I was just asking to mention and discuss the potential for combination effects to occur and affect the results during the screen. Obviously, measuring MIC changes in the clean knock outs in the absence of Kan/Chl confirm the effect of the mutation on Carb/Genta.

Response to Reviewer #1:

The authors have addressed all my comments and I support the publication of the manuscript

We thank the reviewer for the positive and constructive feedback during both revision rounds.

Response to Reviewer #2:

The authors have fully addressed the issues I raised. However, it's not clear whether and how the data will be made publicly available.

We thank the reviewer for the positive and constructive feedback during both revision rounds.

All data is now publicly available:

Illumina sequencing data and transcriptomics data are provided on the EMBL-EBI European Nucleotide Archive (ENA) online repository: ERP159795 <https://www.ebi.ac.uk/ena/browser/view/PRJEB75208> .

Metabolomics data are provided on the MassIVE repository:

MassIVE MSV000094698 for targeted metabolomics
<https://massive.ucsd.edu/ProteoSAFe/dataset.jsp?task=dbf28826aebd4272a0743bf4ce3ae70a>

MassIVE MSV000094699 for untargeted metabolomics
<https://massive.ucsd.edu/ProteoSAFe/dataset.jsp?task=3a3e076019b24a0ca20122bdae93c514>

Response to Reviewer #3:

The authors carefully addressed most of the comments. However, there are two follow-ups detailed below.

We thank the reviewer for the positive and constructive feedback during both revision rounds. And included the two remaining points.

L91 'Although these mutants do not meet the strict definition of resistance, as they do not grow at high drug concentrations, we will refer to them as resistance mutations in the following.'

Better: Although these mutants do not meet the definition of clinical resistance, because they do not grow above the clinical breakpoint concentrations when tested under standard conditions, we will refer to them as resistance mutations for simplicity in the following.

We included the sentence as suggested.

Initial reviewer comment:

CRISPR screen: the authors state in L393 that they added kanamycin and chloramphenicol if the strains harbored pTS040 and pTS041. It is very difficult for readers to understand that those are the plasmids harbored by the strains in the pooled library. This should be clarified. To avoid ambiguity, the authors should clearly state any additional antibiotics contained in each medium for all the different assays described in the methods section. E.g. L515 Did the M9 agar plates contain Kan and Cm? If so, the authors should discuss any bias that might be introduced by the combination effect of Kan or Cm with Carb or Genta. E.g. Yeh et. al showed that beta-lactams (ampicillin) and aminoglycosides (tobramycin) have synergistic effects, while chloramphenicol and aminoglycosides (amikacin) can have antagonistic effects in E. coli.

Reply:

L126 'To avoid any bias from the antibiotics used for plasmid selection (kanamycin and chloramphenicol), we used a control strain containing the same plasmids (pTS040 and pTS041) and cultivated it under the same conditions as the CRISPR library, where kanamycin and chloramphenicol were added to all cultures.'

The provided reply by the authors did not sufficiently exclude nor discuss the possibility of combination effects. This is important because the authors measure relatively small changes in MIC, which could be caused by such combination effects. The provided data is not included in the manuscript and does not determine the combination effect in a clean manner. One possibility would be to determine fractional inhibitory concentration index (FIC_i) (e.g. <https://academic.oup.com/jac/article/79/9/2394/7713050> as a start). However, I was just asking to mention and discuss the potential for combination effects to occur and affect the results during the screen. Obviously, measuring MIC changes in the clean knock outs in the absence of Kan/Chl confirm the effect of the mutation on Carb/Genta.

We apologize that we did not completely solve this point, and agree that the potential of combinatory effects should be included in the manuscript. To address the combinatory effect, we repeated the spot assays of Figure 2 B and D but without kanamycin and chloramphenicol in the agar plates for the PurA^{L75D} and the HemA^{L276Q} mutants (see new Appendix **Figure S4**). We performed the same procedure as described in the method section but washed the cells twice with M9 without antibiotic addition to remove all remaining kanamycin/chloramphenicol from the spots. For carbenicillin, the result is very similar to what we show in Figure 2B when the selection markers are present. The spots for HemA^{L276Q} do not fully grow at the highest concentration tested. As this effect also holds true for the control strain at the lowest concentration, it does not affect our conclusions or further experiments in the paper. The HemA^{L276Q} strain remains resistant towards gentamicin, although on a slightly lower concentration. We added this part in the revised manuscript at line 235:

“To test potential combination effects of carbenicillin or gentamicin with the selection markers kanamycin and chloramphenicol, we made agar dilution assays on plates without the selection markers for the PurA^{L75D} and HemA^{L276Q} mutants. Both mutants showed resistance in the absence of kanamycin and chloramphenicol (Appendix Figure S4), but the susceptibility towards gentamicin was slightly increased for both the control and HemAL276Q strain.”

12th Dec 2024

Manuscript number: MSB-2024-12503RR

Title: Metabolic mutations reduce antibiotic susceptibility of E. coli by pathway-specific bottlenecks

Dear Hannes,

Congratulations on your excellent work! I'm pleased to inform you that your manuscript has been accepted for publication in Molecular Systems Biology. It has been a pleasure working with you to bring it to this stage.

Kind regards,
Jingyi

Jingyi Hou, PhD
Senior Editor
Molecular Systems Biology
